# Neural mechanisms of economic choices in mice

**Masaru Kuwabara, Ningdong Kang, Timothy E Holy, Camillo Padoa-Schioppa***

Department of Neuroscience, Washington University, Saint Louis, United States

**Abstract** Economic choices entail computing and comparing subjective values. Evidence from primates indicates that this behavior relies on the orbitofrontal cortex. Conversely, previous work in rodents provided conflicting results. Here we present a mouse model of economic choice behavior, and we show that the lateral orbital (LO) area is intimately related to the decision process. In the experiments, mice chose between different juices offered in variable amounts. Choice patterns closely resembled those measured in primates. Optogenetic inactivation of LO dramatically disrupted choices by inducing erratic changes of relative value and by increasing choice variability. Neuronal recordings revealed that different groups of cells encoded the values of individual options, the binary choice outcome and the chosen value. These groups match those previously identified in primates, except that the neuronal representation in mice is spatial (in monkeys it is good-based). Our results lay the foundations for a circuit-level analysis of economic decisions.

**\*For correspondence:**
camillo@wustl.edu

**Competing interests:** The authors declare that no competing interests exist.

## Introduction

Economic choice entails computing and comparing the subjective values of different goods. In primates, considerable evidence accumulated in the past 15 years indicates that these operations involve the orbitofrontal cortex (OFC). In humans, OFC neurodegeneration or dysfunction is associated with abnormal decision making (*Barch et al., 2016*; *Bechara et al., 1994*; *Camille et al., 2011*; *Fellows, 2011*; *Hodges, 2001*; *Rahman et al., 1999*; *Waltz and Gold, 2016*; *Yu et al., 2018*). Furthermore, neural signals recorded in OFC during choices correlate with subjective values (*Arana et al., 2003*; *Chaudhry et al., 2009*; *Gottfried et al., 2003*; *Hare et al., 2008*; *Howard et al., 2015*; *Howard and Kahnt, 2017*; *Klein-Flügge et al., 2013*; *Peters and Büchel, 2009*). In non-human primates, neurons in OFC appear intimately involved with the decision process. One series of studies examined head-fixed monkeys choosing between different juices. Relative values were inferred from choices and used to interpret neuronal activity (*Padoa-Schioppa and Assad, 2006*). Different groups of cells in OFC were found to encode the value of individual goods (*offer value*), the binary choice outcome (*chosen juice*) and the *chosen value*. These variables capture both the input (*offer value*) and the output (*chosen juice*, *chosen value*) of the choice process, suggesting that the groups of neurons identified in OFC constitute the building blocks of a decision circuit. Supporting this hypothesis, trial-by-trial fluctuation in neuronal activity correlates with choice variability (*Padoa-Schioppa, 2013*), the activity dynamics of neuronal populations in OFC reflects an internal deliberation (*Rich and Wallis, 2016*), and suitable electrical stimulation of OFC biases or disrupts economic decisions (*Ballesta and Padoa-Schioppa, 2019*). Complementing these empirical findings, computational work showed that neural networks whose units resemble the groups of cells identified in OFC can generate binary choices (*Friedrich and Lengyel, 2016*; *Rustichini and Padoa-Schioppa, 2015*; *Solway and Botvinick, 2012*; *Song et al., 2017*; *Zhang et al., 2018*). In summary, primate studies consistently implicate the OFC in the generation of economic decisions.

In comparison, the picture emerging from research in rodents is more heterogenous. Anatomically, the primate OFC (areas 13/11) appears homologous to the rodent lateral orbital (LO) area

(*Ongür and Price, 2000*). Consistent with this understanding, LO lesions in rodents (*Gallagher et al., 1999*; *Gremel and Costa, 2013*) and OFC lesions in primates (*Izquierdo et al., 2004*; *Reber et al., 2017*; *Rudebeck and Murray, 2011*; *West et al., 2011*) induce similar effects on goal-directed behavior under reinforcer devaluation. A series of studies found evidence consistent with a neuronal representation of value and other decision variables in LO (*Feierstein et al., 2006*; *Hirokawa et al., 2019*; *Roitman and Roitman, 2010*; *Sul et al., 2010*; *van Duuren et al., 2008*; *van Duuren et al., 2009*; *Zhou et al., 2019*). For example, van Duuren and colleagues showed that neurons in LO reflects both the magnitude and the probability of upcoming rewards. As a caveat, these experiments did not include real economic choices or trade-offs between competing dimensions. Other experiments showed that LO lesions disrupted choices, but some results have been contradictory (*Mobini et al., 2002*; *Rudebeck et al., 2006*; *Winstanley et al., 2004*). Concurrently, several studies cast doubt on the notion that neurons in LO encode economic values (*McDannald et al., 2014*; *Roesch et al., 2006*), or that values represented in this area affect economic decisions (*Miller et al., 2018*; *Stott and Redish, 2014*). Again, an important caveat is that these experiments did not include any dimensional trade-off. However, in one recent study, rats performed a juice choice task similar to that used in primates (*Gardner et al., 2017*). Surprisingly, the authors found that optogenetic inactivation of LO did not disrupt choices. This result raised the possibility that economic decisions operate in fundamentally different ways in rodents and primates, or perhaps that seemingly minor differences in task design induce different decision mechanisms (see Discussion).

Assessing whether economic decisions in primates and rodents are supported by homologous brain areas is valuable from a comparative perspective. More importantly, many aspects of the decision circuit identified in non-human primates and discussed above remains poorly understood. For example, it is unclear whether different groups of neurons identified in relation to behavior correspond to different anatomical cell types, or whether these neurons reside in different cortical layers. It is also unclear whether these groups of neurons are differentially connected with each other and with other brain regions. Addressing these questions in monkeys is technically difficult. However, these issues can in principle be addressed using genetic tools available in mice. For this reason, establishing a credible mouse model to investigate this decision circuit is potentially transformative.

Here we present such model. More specifically, the study describes three primary results. First, we developed a mouse version of the juice choice task. In the experiments, head-fixed mice chose between two liquid rewards (juices) offered in variable amounts. Each juice was associated with an odor, and the odor concentration indicated the juice quantity. In each trial, the two odors indicating the offers were presented simultaneously from two directions, and the animal revealed its choice by licking one of two liquid spouts. Apart from using olfactory stimuli instead of visual stimuli, this task closely resembled that used for monkeys (*Padoa-Schioppa and Assad, 2006*). Mice learned the task rapidly and generally exhibited choice patterns similar to those measured in primates. Second, we used optogenetics to examine the effects of LO inactivation on choices. Inactivation was induced by optically activating GABAergic interneurons expressing channel-rhodopsin2 (ChR2). LO inactivation severely disrupted the decision process, in the sense that it altered the relative values of the two juices and consistently increased choice variability. Hence, economic decisions in mice require area LO. Third, we recorded and analyzed the spiking activity of neurons in LO. Different groups of cells encoded the value of individual offers, the binary choice outcome and the chosen value. In this respect, neuronal responses in LO closely resembled those recorded in the primate OFC. The main difference between the two species was that neurons in LO represented offers and values in a spatial frame of reference, whereas the representation in the primate OFC was good-based (*Padoa-Schioppa and Assad, 2006*). Furthermore, neurons in LO preferentially encoded the value offered on the ipsi-lateral side, suggesting that economic decisions in mice ultimately involved a competition between the two hemispheres. These results address outstanding questions and establish a new and powerful approach to study the neural circuit underlying economic decisions.

## Results

### Economic choice behavior in mice

We developed a behavioral paradigm similar to that previously used for monkeys. In essence, we let mice choose between two liquid rewards (juices) offered in variable amounts. During the experiments, mice were head-fixed, and two liquid spouts were placed close to their mouth, on the two sides. For each juice, the offered quantity varied from trial to trial. A key aspect of the experiment was to effectively communicate to the animal the two options available on any given trial. We represented offers using olfactory stimuli because mice can easily learn to make subtle olfactory discriminations (*Wachowiak et al., 2009*). We used odor identity to represent a particular juice type, and odor concentration to represent juice quantity. In each trial, the animal was presented with two odors from the two directions (left, right). The odors, representing the offers, were presented for 2.8 s (offer period), at the end of which the animal heard an auditory 'go' signal. The animal indicated its choice by licking one of the two liquid spouts, and the corresponding juice was delivered immediately thereafter (*Figure 1a*). Throughout the experiments, we used five odor concentrations, corresponding to five quantity levels for each juice. Juice quantities varied on a linear scale, while odor concentrations varied roughly linearly on a log2 scale. Juice quantities and left/right positions varied pseudo-randomly from trial to trial (see Materials and methods).

Animals' choices reliably presented a quality-quantity trade-off. *Figure 1b–f* illustrate the behavior observed in five representative sessions. We refer to the two juices as A and B, with A preferred. If the two juices were offered in equal amounts (1B:1A), the animal would reliably choose juice A (by definition). However, if juice B was offered in sufficiently large amount against 1A, the animal chose B. For example, in *Figure 1f*, the mouse was roughly indifferent between 1A and 2B. For a quantitative analysis, we ran a logistic regression (see Materials and methods, *Equation 1*). The logistic fit provided measures for the relative value of the two juices (ρ) and for the sigmoid steepness (η), which is inversely related to choice variability. For example, for the session in *Figure 1f*, we measured ρ = 2.2 and η = 2.1.

The present study is based on 19 mice (see Materials and methods) and a total of 335 sessions. Animals typically performed for 250–400 trials per session. *Figure 2* summarizes the whole behavioral data set (excluding trials with optogenetic inactivation; see below). Across sessions and across mice, relative values varied broadly (mean(ρ)=2.39; *Figure 2a*). Similarly, the sigmoid steepness varied from session to session (mean(η)=1.19; *Figure 2b*). In general, ρ >1 implies that choices are based on values that integrate juice type and juice quantity. In this sense, mice reliably presented non-trivial choice patterns, as previously observed for monkeys (*Padoa-Schioppa and Assad, 2006*). Conversely, the sigmoid steepness measured in mice was generally lower (higher choice variability) than that recorded in monkeys.

In most of our experiments, higher odor concentrations represented larger juice amounts. One concern was whether low odor concentrations were hard to discern, and whether choices were ultimately dictated by perceptual ambiguity. One argument against this hypothesis follows from the fact that we observed similar choice patterns independently of the pairing between odors and juice type. For an additional control, we trained two animals in a 'flipped' version of the task, in which higher odor concentrations represented smaller juice amounts. These two mice learned the task and eventually performed very similarly to the other animals (*Figure 1g*). More specifically, the relative value and sigmoid steepness measured for these two mice was statistically indistinguishable from those measured for the other 17 mice (p=0.12 and 0.15 for ρ and η, respectively; *t* test; *Figure 2*, asterisks).

### Inactivation of area LO disrupts economic decisions

To shed light on the role of LO in economic decisions, we examined how inactivation of this area affects choices. In a series of experiments, we inactivated LO by optically activating GABAergic interneurons expressing ChR2. We chose this protocol because exciting inhibitory cells often induces a more complete shut-down of projection neurons (*Wiegert et al., 2017*; *Zhao et al., 2011*).

First, we tested the effects of LO inactivation in two VGAT-ChR2 mice, which (in principle) express ChR2 in all GABAergic cells throughout the brain. In each session, inactivation and control trials were pseudo-randomly interleaved. Inactivation was induced by shining blue light bilaterally in area

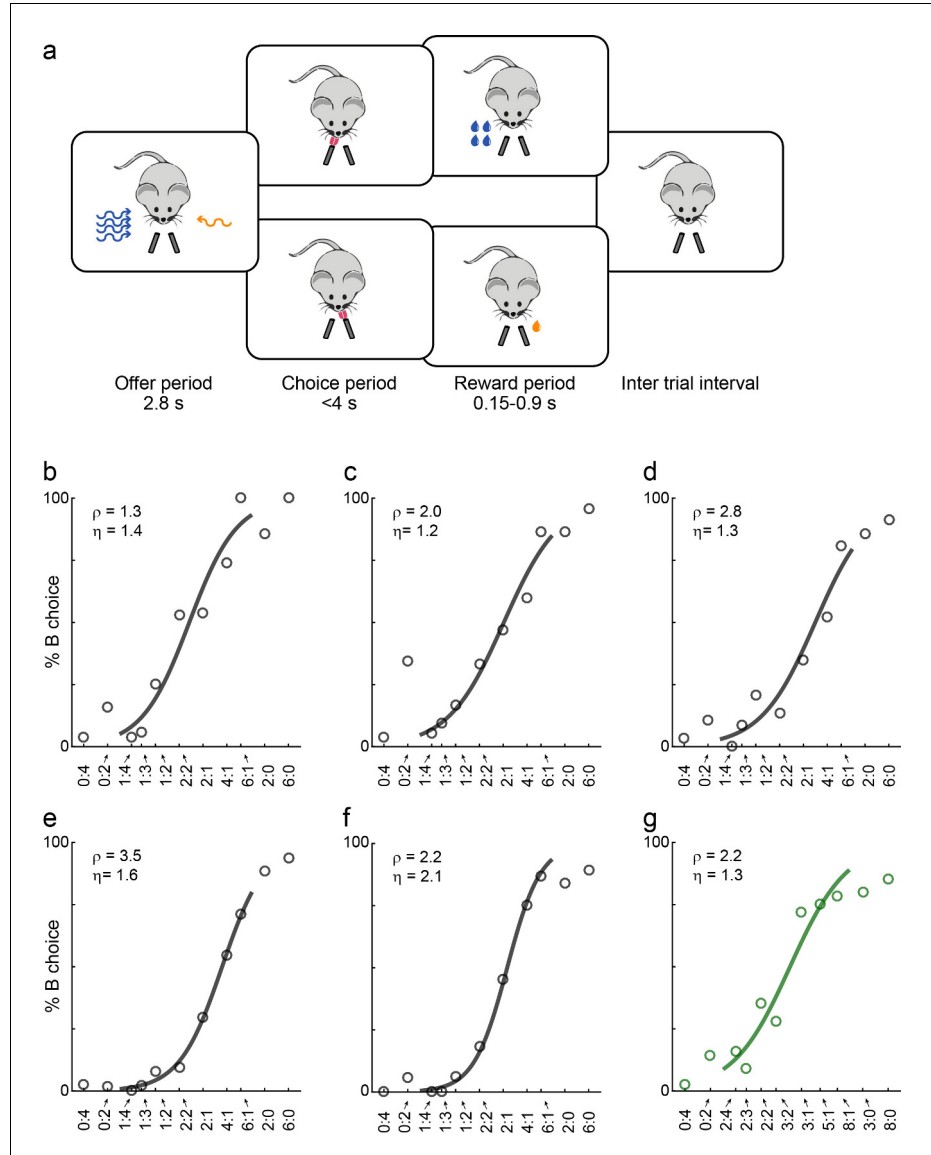

**Figure 1.** Economic choices in mice. (a) Task diagram. In the experiments, mice chose between two liquid rewards (juices, schematized by the two lick tubes) offered in variable amounts. For each offer, the juice identity (water or sugar water) was represented by the odor identity, and the juice quantity was represented by the odor concentration. (b–f) Choice pattern, example sessions. In each panel, the x-axis represents different offer types in $\log(q_B/q_A)$; the y-axis represents the percent of trials in which the animal chose juice B. Data points are averaged across trials, and the sigmoid is obtained from a logistic regression (Materials and methods, *Equation 1*). Each panel indicates the relative value (ρ) and the sigmoid steepness (η). The five sessions shown here are from mice M56, M55, M47, M48 and M49, respectively. Forced choices are shown (including error trials) but not included in logistic regressions. (g) Example session of a mouse performing the flipped version of the task (M53).

LO during the offer and choice periods (see Materials and methods). This manipulation consistently disrupted choices. Across sessions, LO inactivation altered the relative value in a seemingly erratic way, and consistently reduced the sigmoid steepness (i.e., it increased choice variability). These effects were quantified with logistic analyses (Materials and methods, *Equation 5*), which provided measures of relative value ($\rho_{stim\ OFF}$, $\rho_{stim\ ON}$) and sigmoid steepness ($\eta_{stim\ OFF}$, $\eta_{stim\ ON}$). In both mice, the distribution of relative values measured across sessions was significantly broader under LO inactivation than in normal conditions (both $p < 10^{-5}$, F test for equality of variance; *Figure 3a*;

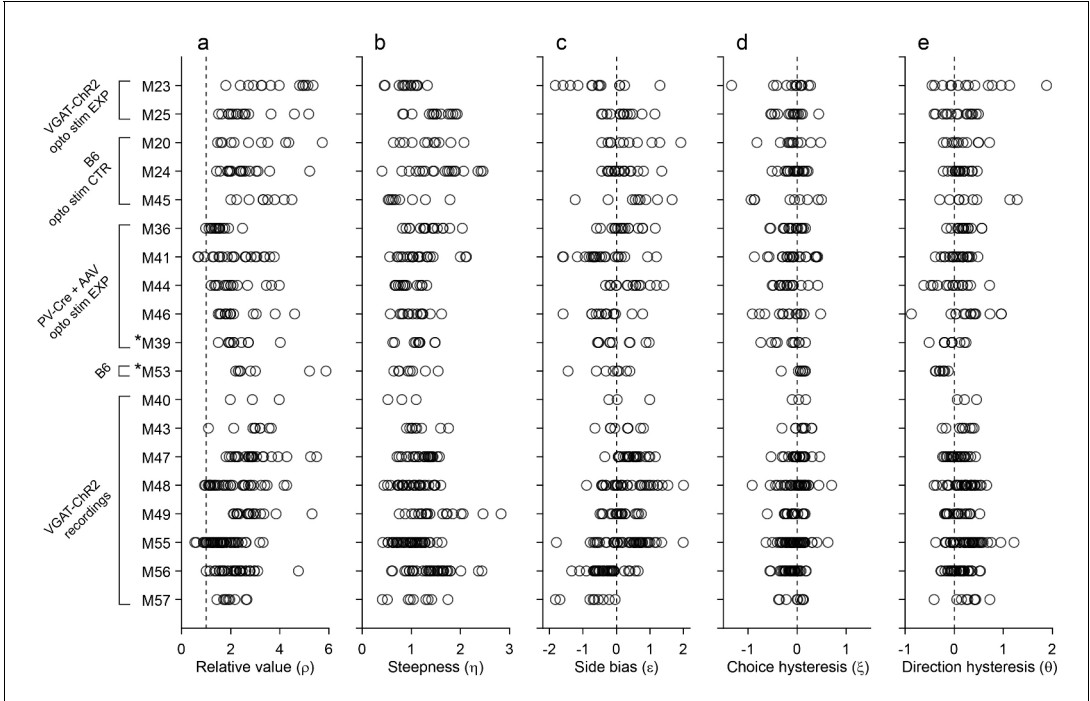

**Figure 2.** Behavior, population analysis. The figure illustrates the entire behavioral data set (19 mice). Each panel illustrates one behavioral measure, each row represents one mouse, and each data point represents one session. Relative value and steepness were obtained from *Equation 1*; side bias, choice hysteresis and direction hysteresis were obtained from *Equations 2-4*. Labels on the left indicate for each mouse the strain and the relevant experiment. Asterisks indicate the two mice (M39 and M53) trained in the flipped version of the task. Importantly, the measures obtained for these mice were comparable to those obtained for the other animals. For mice participating in optical inactivation experiments, only trials without stimulation (stimOFF) were included in this figure. (a) Relative value. Averaging across sessions and across animals, we obtained mean($\rho$)=2.39 ± 1.01 (mean ± SD). (b) Steepness. Averaging across sessions and across animals, we obtained mean($\eta$)=1.19 ± 0.41 (mean ± SD). (c) Side bias. On any given session, choices could present a side bias, although the direction changed across sessions and across animals. Seven animals presented a consistent bias (p<0.01, *t* test). Of them, four preferred right and three preferred left, indicating that side biases were not imposed by the experimental apparatus. Averaged across the population, the side bias was rather modest (mean($\epsilon$)=0.08 ± 0.04, mean ± SEM). (d) Choice hysteresis. Only one animal presented a consistent effect (p<0.01, *t* test). Averaged across the population, choice hysteresis was modest (mean($\xi$) = –0.08 ± 0.01, mean ± SEM). (e) Direction hysteresis. Six animals presented a consistent effect (p<0.01, *t* test). For one of them the effect was negative. Averaged across the population, direction hysteresis was significant but fairly modest ($\theta$ = 0.15 ± 0.02, mean ± SEM).

*Figure 4ab*). Conversely, the sigmoid steepness was consistently lower under LO inactivation than in normal conditions (both p<10$^{-4}$, paired *t* test; *Figure 3a*; *Figure 5ab*).

These results suggested that decisions relied on the neuronal activity in area LO. One concern was the extent of the inactivated region. Since VGAT-ChR2 mice express ChR2 in interneurons throughout the brain, and since the light spreads to some extent through the brain, the behavioral effects described above could in principle be due to the inactivation of neighboring areas such as the olfactory bulb or the motor cortex. To address this issue, we conducted a second experiment. In this case, we tested PV-Cre mice, in which we injected AAV-DIO-ChR2 specifically in area LO (*Figure 3g*). PV neurons are inhibitory cells that target the perisomatic domain of local pyramidal cells (*Tremblay et al., 2016*). In this preparation, optical stimulation activated PV neurons exclusively in the injected region, resulting in specific inactivation of area LO. We repeated optogenetic inactivation experiments in five PV-Cre mice. Confirming our initial results, in each mouse the distribution of relative values was significantly broader under LO inactivation than in normal conditions (all p<0.002, *F* test for equality of variance; *Figure 3*; *Figure 4c–h*). Furthermore, in each mouse, the sigmoid steepness was consistently lower under LO inactivation than in normal conditions (all p<10$^{-3}$, paired *t* test; *Figure 3*; *Figure 5c–h*).

Another concern was the fact that area LO is not far from the eyes. Thus the blue light shone during stimulation trials might be seen by the mouse. In principle, such visual input could distract the animal or otherwise interfere with its choosing. To address this issue, we conducted a control

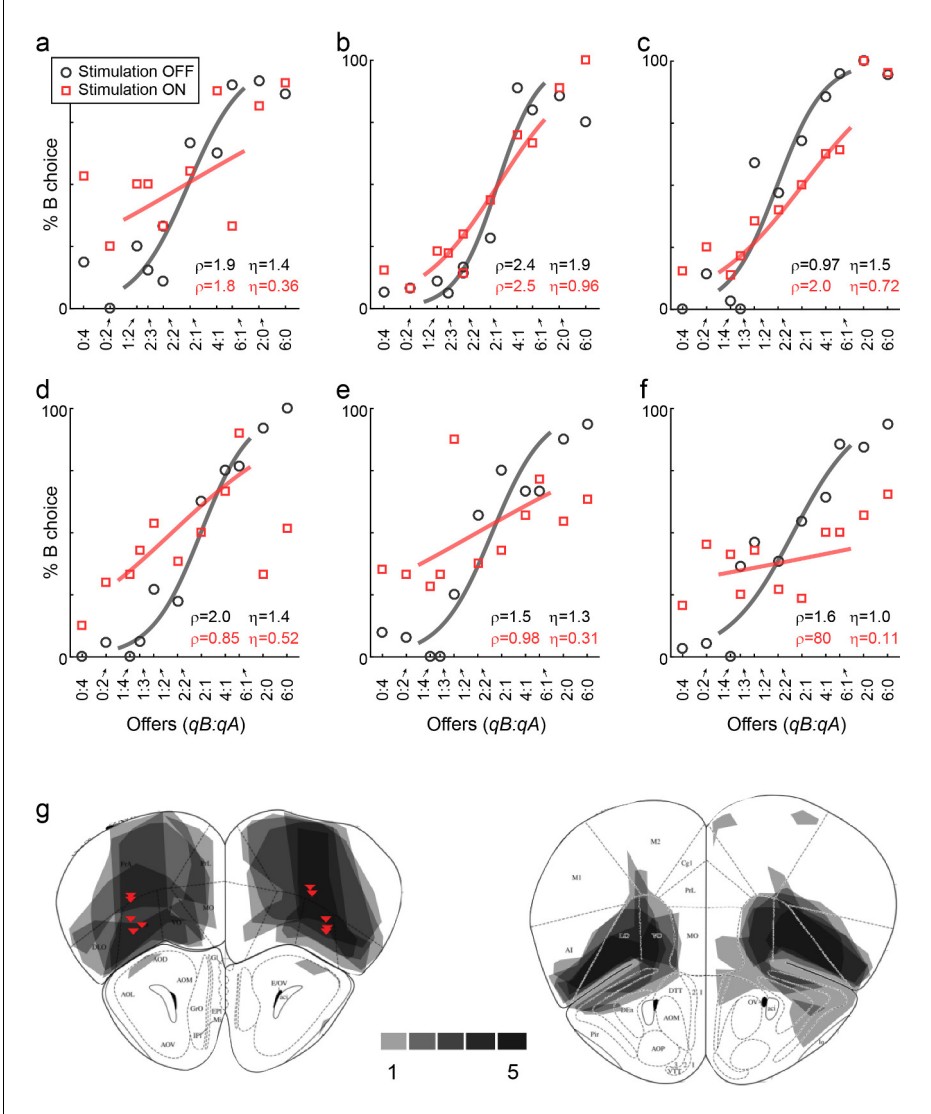

**Figure 3.** LO inactivation disrupts economic decisions, example sessions. (**a**) Effect of LO inactivation measured in a VGAT-ChR2 mouse. (**b–f**) Effect measured in PV-Cre+AAV-DIO-ChR2 mice. Each panel illustrates one session. Red data points represent trials in which area LO was inactivated through optical stimulation of GABAergic interneurons (stimulation ON). Black data points represent normal conditions (stimulation OFF). Each panel indicates the relative value ($\rho$) and the sigmoid steepness ($\eta$) measured in each condition. As a result of LO inactivation, $\rho$ varied in a seemingly erratic way, while $\eta$ consistently increased (shallower sigmoids). The six sessions shown here are from mice M25, M24, M36, M46, M41 and M41, respectively. (**g**) Histological analysis of adeno-associated virus (AAV) injections. The infected area for five mice are overlaid in gray scale (mouse atlas, sections A2.68 and A2.22) (**Franklin and Paxinos, 2013**). Red triangles indicate the locations of the optical fiber tips.

experiment, in which we repeated the same light stimulation protocol of area LO in three mice that did not express ChR2 (see Materials and methods). In this case, the stimulation did not affect choices in any appreciable way. In particular, the distributions of relative values under stimulation were indistinguishable from those measured without stimulation (all p>0.2, *F* test for equality of variance; *Figure 4i–l*). Similarly, the sigmoid steepness was generally indistinguishable from that measured without stimulation (all p>0.09, paired *t* test; *Figure 5i–l*).

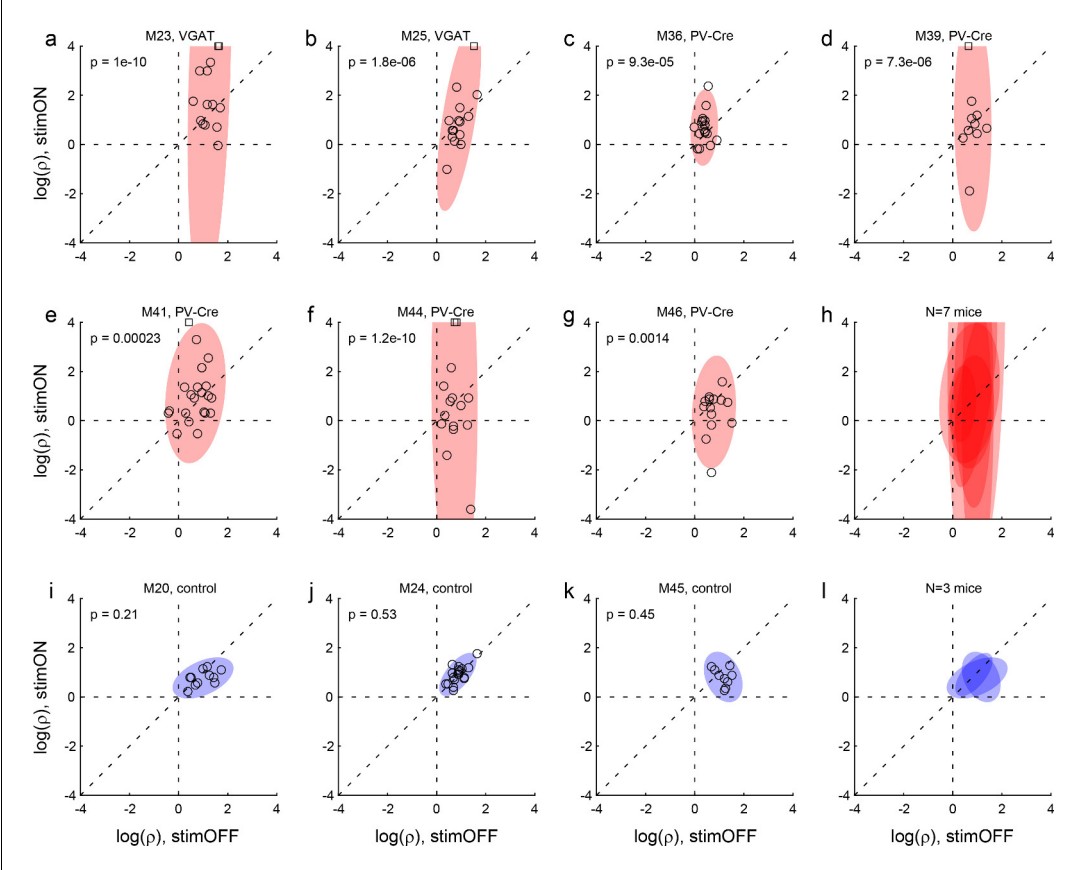

**Figure 4.** Effects of LO inactivation, relative value. (**a, b**) Results for two VGAT-ChR2 mice. Each panel illustrates the results for one animal. The x- and y-axes represent the log relative value, log(ρ), measured under normal conditions (stimOFF) and under LO inactivation (stimON), respectively. Each data point represents one session. Outliers are clipped to the axes and indicated with a square. Shaded ellipses represent the 90% confidence interval for the corresponding distribution. Under normal conditions, log(ρ)>0 and the distribution across sessions was relatively narrow in both animals. As a result of LO inactivation, log(ρ) increased or decreased, in way that appeared erratic. The resulting distribution for log(ρ) was much wider (ellipses are elongated along the vertical axis). This effect was highly significant in both animals. The p value indicated in each panel is from an *F* test for equality of variance. (**c–g**) Results for five PV-Cre + AAV-DIO-ChR2 mice. The results closely resemble those in panels (**ab**). (**h**) Combined results for seven experimental mice. The ellipses illustrated in panels (**a–g**) are superimposed here. (**i–k**) Results for three control mice (see Results). Same convention as in (**a**). In this case, the log(ρ) measured under optical stimulation was generally indistinguishable from that measured under normal conditions. (**l**) Combined results for three control mice.

## Reversion to stereotyped behavior

We investigated more specifically possible ways in which LO inactivation might disrupt decisions. In general, choices can be influenced by other factors besides the juice types and juice quantities. For example, previous work in unrestrained pigeons and rats found consistent side biases (*Kagel et al., 1995*). Similarly, other things equal, monkeys tend to choose on any given trial the same juice chosen in the previous trial (choice hysteresis) (*Padoa-Schioppa, 2013*). Here we examined three possible sources of choice biases related to the spatial configuration of the offers (side bias) and to the outcome of the previous trial (choice hysteresis, direction hysteresis). Each effect was examined separately and with a logistic analysis (see Materials and methods, *Equations 2-4*).

We first examined choice biases under normal conditions. *Figure 6* illustrates the results for one representative session. In this session, the animal presented a sizeable side bias (ε = –0.66; *Figure 6b*), negligible choice hysteresis (ξ = 0.13; *Figure 6c*), and some direction hysteresis (θ = 0.42; *Figure 6d*). Similar results held across sessions and across animals. In any given session, mice could present some bias favoring either the left or the right option. However, the direction of the side bias varied across sessions and across animals (*Figure 2c*), indicating that side biases did not reflect asymmetry in the experimental apparatus. Choice hysteresis was generally low

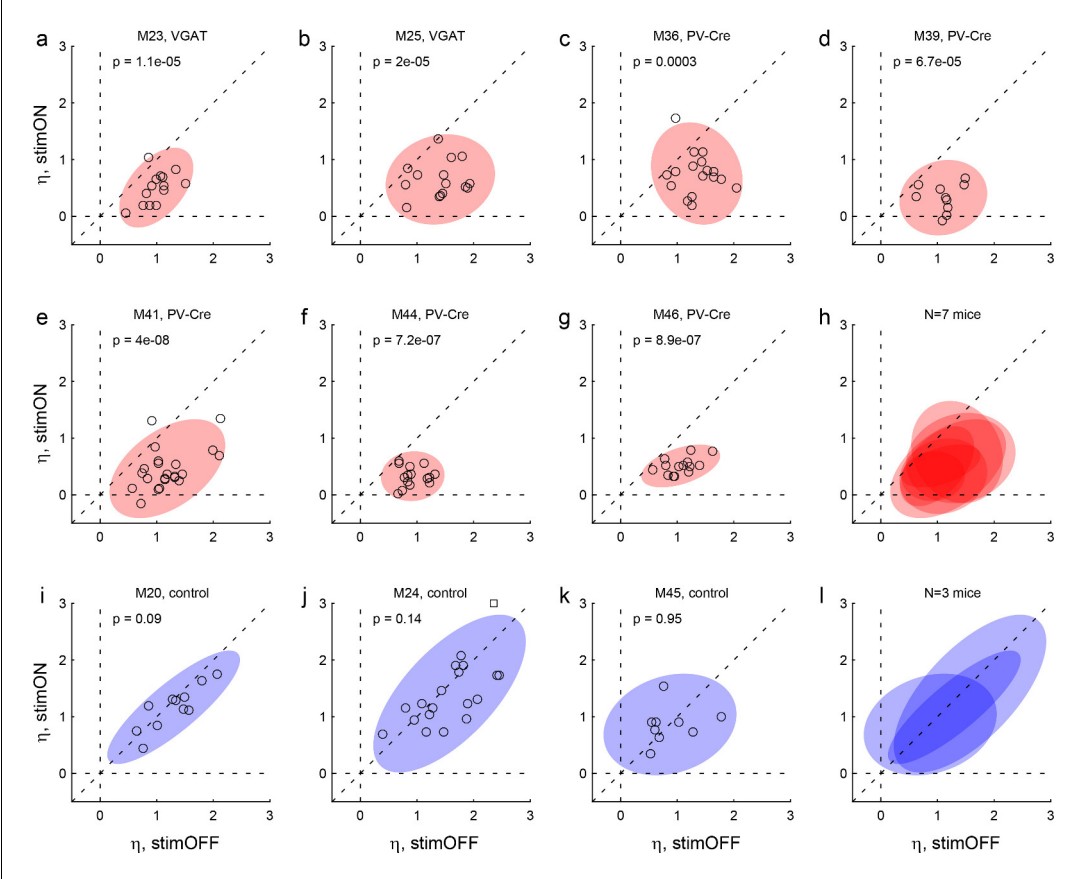

**Figure 5.** Effects of LO inactivation, sigmoid steepness. (a, b) Results for two VGAT-ChR2 mice. The x- and y-axes represent the sigmoid steepness (η) measured under normal conditions (stimOFF) and under LO inactivation (stimON), respectively. Each data point represents one session and shaded ellipses represent the 90% confidence interval for the corresponding distribution. Under normal conditions, η varied somewhat from session to session. Independently of the measure obtained under normal conditions, LO inactivation consistently reduced the sigmoid steepness (data points lie below the identity line). In other words, LO inactivation increased choice variability (choices became more noisy). This effect was highly significant in both animals. The p value indicated in each panel is from a paired *t* test. (c–g) Results for five PV-Cre + AAV-DIO-ChR2 mice. The results closely resemble those in panels (ab). (h) Combined results for seven experimental mice. (i–k) Results for three control mice. For these animals, the steepness measured under optical stimulation was generally indistinguishable from that measured under normal conditions (data points lie on the identity line). (l) Combined results for three control mice.

(*Figure 2d*), and mice presented a small but consistent direction hysteresis (*Figure 2e*). In summary, under normal conditions, choice biases were relatively modest, as choices were dominated by the trade-off between juice type and juice amount.

We next examined how LO inactivation affected choice biases. We found that optical stimulation significantly increased the side bias in 5 of 7 mice (all p<0.05, *F* test for equality of variance) – an effect not observed in any of the control animals (*Figure 6—figure supplement 1*). Similarly, LO inactivation significantly increased choice hysteresis in 5 of 7 mice (all p<0.01, *F* test for equality of variance; *Figure 6—figure supplement 2*). Finally, LO inactivation significantly increased direction hysteresis in 5 of 7 animals (all p<0.05, *F* test for equality of variance; *Figure 6—figure supplement 3*). Interestingly in some sessions, choice hysteresis (ξ) and direction hysteresis (θ) became negative under optical stimulation, indicating that LO inactivation induced choice alternation rather than choice repetition.

In summary, LO inactivation reduced performance by introducing a variety of choice biases. Normally, economic decisions take place through the computation and comparison of subjective values. Absent LO, animals seem to revert to stereotyped behaviors, whereby choices are dictated by the spatial location (side bias) or by the recent history (hysteresis).

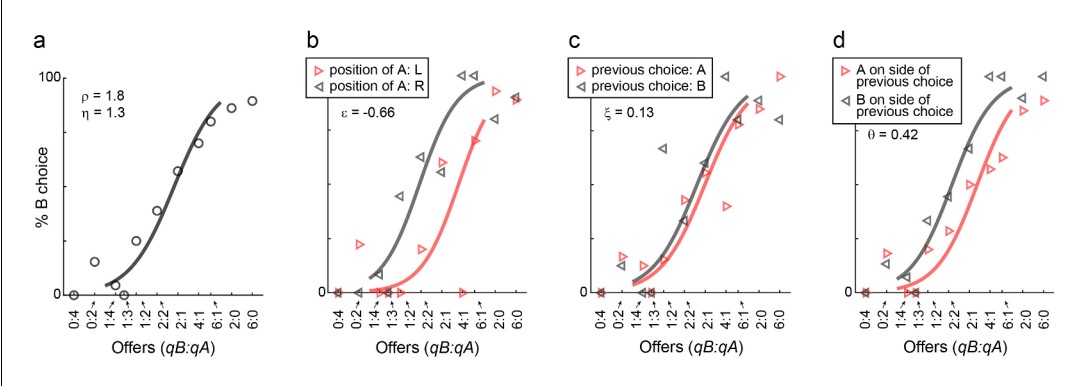

**Figure 6.** Choice biases, example session. (a) Example session (M57), same format as in *Figure 1*. (b–d) Choice biases. For the session in panel (a), the three panels illustrate the side bias (b), the choice hysteresis (c) and the direction hysteresis (d). In (b), trials were divided depending on the position of juice A (left or right). In (c), trials were divided depending on the juice chosen on the previous trial (A or B). In (d), trials were divided depending on whether juice A was offered on the side chosen in the previous trial or on the other side. In each panel, data points are averages across trials. Sigmoids were obtained from logistic regressions (Materials and methods, *Equations 2-4)* and each panel indicates the corresponding parameter (ε, ξ, θ). The online version of this article includes the following figure supplement(s) for figure 6:

**Figure supplement 1.** Effects of LO inactivation, side bias.
**Figure supplement 2.** Effects of LO inactivation, choice hysteresis.
**Figure supplement 3.** Effects of LO inactivation, direction hysteresis.

## Choice deficits reflect LO inactivation

Despite the effort to confine injections to area LO, viral infections partly spread to neighboring brain areas (*Figure 3g*). Thus one concern was whether the behavioral deficits described above were due to interference with olfactory processes caused by inactivation of piriform cortex and/or the olfactory bulb. More generally, we sought to assess whether choice deficits reflected inactivation of some area different from LO. To address this issue, we examined the 5 PV-Cre mice and we conducted a regression analysis relating choice deficits to viral infection in different brain areas. We proceeded as follows.

First, using histological images (*Figure 3g*), we quantified for each mouse the extent of viral infection by brain region. Across five animals, some degree of infection was measured in 12 distinct areas, namely FrA, PrL, MO, VO, LO, DLO, OB, M1, M2, CgI, AI and EX (*Franklin and Paxinos, 2013*). For each animal and each area, we quantified the number of stained pixels and interpreted it as a measure of the infected volume. (Some areas were infected only in a subset of mice. Conversely, for areas PrL, MO, VO and LO, we had measures from two AP coordinates; *Figure 3g*) We thus obtained a matrix where rows and columns represented mice and areas, respectively, and each element indicated the infected volume.

Second, we noted that the primary effect of the optogenetic manipulation on choices was to decrease the sigmoid steepness η (*Figure 5*). For each session, we quantified this effect by defining the steepness stimulation index (SSI):

$$\text{SSI} = 2^*(\eta_{\text{stim ON}} - \eta_{\text{stim OFF}})/(\eta_{\text{stim ON}} + \eta_{\text{stim OFF}})$$

For each mouse, the choice deficit was defined as CD = –mean(SSI), where the average was across sessions and the minus sign made it so that larger CDs corresponded to more severe behavioral impairments.

Third, we performed a LASSO regression (*Tibshirani, 1996*) of CD against the infected volume in each area. We used this procedure because the number of areas exceeded the number of measurements (i.e., the number of animals). The LASSO regression is parsimonious in the sense that it limits the number of covariates in the final model. If inactivation of a particular area reduced choice performance, the regression coefficient obtained for that area would be positive. Of all the areas included in the analysis, only three presented a non-zero coefficient, and only one of them, namely LO, presented a positive sign. In other words, only for area LO did the choice deficit and the volume of

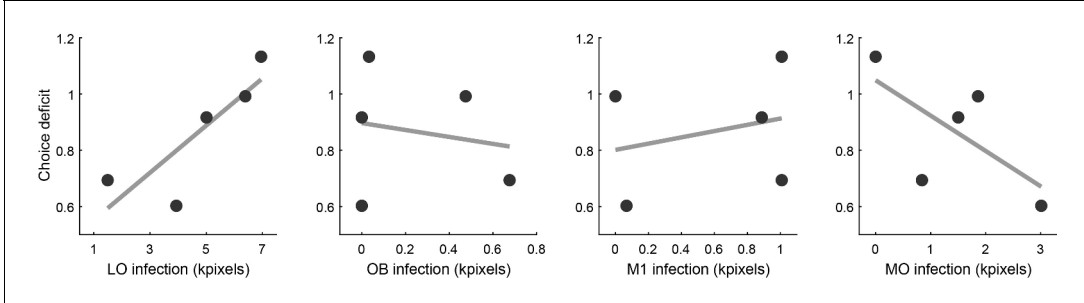

**Figure 7.** Relation between choice deficit and infection in four brain areas. Panels from left to right illustrate the results obtained for area LO, the olfactory bulb (OB), primary motor cortex (M1) and the medial orbital area (MO). In each panel, the x-axis represents the infection size (i.e., a proxy for the level of inactivation under optical stimulation). The y-axis represents the choice deficit, with higher values corresponding to more severe impairments. Each data point represents one animal. Only for area LO did infection size and choice deficit correlate in the predicted direction. For OB and M1 there was no correlation. For MO, the correlation was in the opposite direction.

infection correlate in the predicted direction (**Figure 7**). This result strongly supported our interpretation.

Another concern was whether choice deficits illustrated in **Figure 5** reflected some motor impairment, perhaps due to inactivation of area M1. The regression analysis described above argued against this hypothesis. For additional scrutiny, we examined the licking activity of the same PV-Cre mice. In a nutshell, we found that optical stimulation affected licking in a modest, but statistically significant way. Most importantly, optically induced choice deficits were not correlated with this effect on licking. **Figure 8** illustrates our results. **Figure 8a** depicts the licking activity recorded in a representative session. The licking frequency increased during odor presentation and peaked shortly after the go signal. Considering the time window 1.5–4.5 s after the trial start, under normal conditions (stim OFF), the animal made an average of 12.9 licks/trial. Under optical stimulation (stim ON), licking decreased to an average of 11.0 licks/trial. To quantify this effect, we defined the lick stimulation index (LSI):

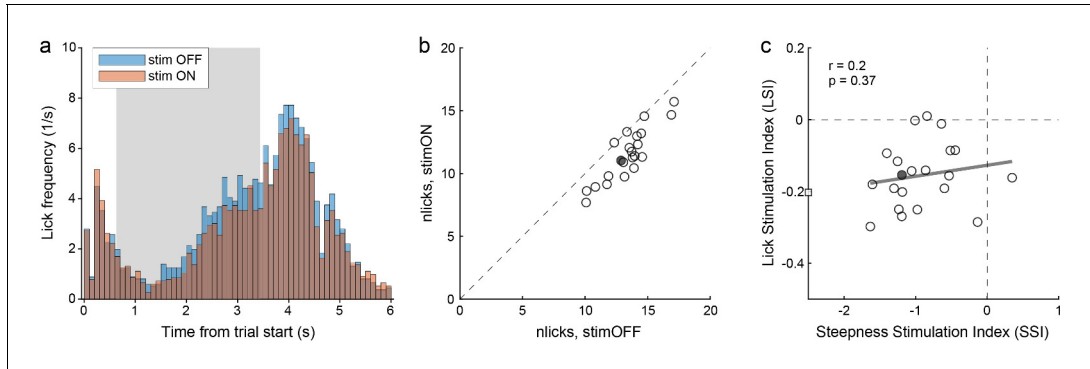

**Figure 8.** Choice deficits were not caused by motor impairments. (a) Licking activity in one representative session (from mouse M41). The plot illustrates the licking frequency (y-axis) over the course of the trial under normal conditions (stim OFF) and under optical stimulation (stim ON). The gray shade highlights the stimulation period. All trials were included in the figure. Notably, the licking frequency decreased somewhat under stimulation. Indeed, for this session we measured LSI = –0.15. (b) Effects of optical stimulation on licking across sessions (mouse M41). The axes represent the average number of licks measured in stim OFF trials (x-axis) and in stim ON trials (y-axis). Measures were obtained from the time window 1.5–4.5 s after the trial start. Each data point represents one session, and the filled data point indicates the session in panel (a). Stimulation reduced licking consistently but modestly. For this animal, we measured mean(LSI) = –0.16. (c) Relation between choice deficits and motor effects (mouse M41). The axes represent the primary effect of optical stimulation on choice (SSI, x-axis) and that on licking (LSI, y-axis). Each data point represents one session, the filled data point represents the session in panel (a), and the square represents an outlier. The gray line was obtained from a linear regression (excluding the outlier). LSI and SSI were not correlated across sessions (r = 0.2, p=0.37). Similarly, we did not find any correlation between LSI and SSI in any of the other 4 PV-Cre mice (all r < 0.2, all p>0.35).

$$LSI = 2*(nlicks_{stimON} - nlicks_{stimOFF})\,/\,(nlicks_{stimON} + nlicks_{stimOFF})$$

where nlicks indicate the average number of licks per trial. LSI < 0 indicated that the manipulation reduced licking, and for the session in *Figure 8a* we obtained LSI = –0.15. An analysis across sessions confirmed that optical stimulation affected licking. Specifically, in four animals we consistently measured LSI < 0 (all p<0.03, t test; *Figure 8b*); in one animal, our measures were statistically indistinguishable from LSI = 0 (p=0.43, t test). On average across sessions and mice, we measured mean (LSI)=–0.08. In contrast, for each of the three control mice, our measures were statistically indistinguishable from LSI = 0 (all p>0.25, t test); averaging across sessions and mice, we obtained mean (LSI)=–0.01.

The key question was whether choice deficits induced by optical stimulation were ultimately caused by motor impairments. If so, the decrease in sigmoid steepness, quantified by SSI, should correlate with the effect on licking, quantified by LSI. We thus examined the relation between SSI and LSI across sessions, separately for each mouse. For mouse M41, the two measures were not correlated (r = 0.2, p=0.37; *Figure 8c*). Similarly, SSI and LSI were not correlated for any of the PV-Cre mice (all r ≤ 0.2, all p>0.35). Thus we concluded that choice deficits induced by the optical stimulation were caused by specific interference with the decision process, as opposed to a generic motor impairment.

## Neuronal activity in area LO during economic decisions

The behavioral effects observed under LO inactivation reveal that this area is necessary for economic decisions. In another set of experiments, we examined the spiking activity of neurons in LO while mice performed the choice task. One of our aims was to compare the results to those previously obtained for central OFC in monkeys. We recorded the activity of 717 cells from eight mice. Of these cells, 197 and 520 were from left and right hemispheres, respectively. We pooled data from different mice and analyzed them similarly to how we analyzed data from monkeys (see Materials and methods). Specifically, we defined five time windows aligned with the beginning of offer presentation and with juice delivery (see Materials and methods). A preliminary assessment indicated that neurons in LO were modulated by the spatial contingencies of the choice task. In the analysis, an 'offer type' was defined by two quantities of juices A and B; a 'trial type' was defined by an offer type, a spatial configuration of the offers, and a choice. For each time window and for each trial type, we averaged spike counts across trials. A 'neuronal response' was defined as the firing rate of one cell in one time window, as a function of the trial type.

*Figure 9* illustrates the activity of six example cells. The response in *Figure 9a* increased as a function of the value offered on the left, independently of whether the juice offered on the left was A or B, and independently of the animal's choice. This neuron was recorded in the left hemisphere. Thus the response seemed to encode the variable *offer value ipsi*. Similarly, the response in *Figure 9b*, recorded in the right hemisphere, seemed to encode the variable *offer value ipsi*. The response in *Figure 9c* was nearly binary. The firing rate was high when the animal chose the offer on the right, and it was low when the animal chose the offer on the left, independent of the chosen juice and the chosen quantity. Thus the response seemed to encode the variable *chosen side*. *Figure 9d* shows another neuronal response encoding the *chosen side*. The response in *Figure 9e* seemed to encode the variable *chosen value*. Its activity increased as a function of the value chosen by the animal, independent of the juice type or the chosen side. Finally, the response in *Figure 9f* seemed to encode the variable *position of A*. Its activity was roughly binary – high when juice A was offered on the left and low when juice A was offered on the right, independent of the quantity and on the animal's choice.

For a quantitative analysis of the whole data set, we proceeded in steps. First, the activity of each neuron in each time window was examined with a 3-way ANOVA (factors: offer type ×position of A × chosen side). This analysis confirmed that many cells were modulated by the offer type, the spatial configuration of the offers, and/or the movement direction (*Table 1*). We also conducted a 1-way ANOVA with factor trial type (which recapitulates information about the offers, their spatial locations and the chosen side). We imposed a significance threshold p<0.001. In total, 565 responses from 301 cells satisfied this criterion and were included in subsequent analyses.

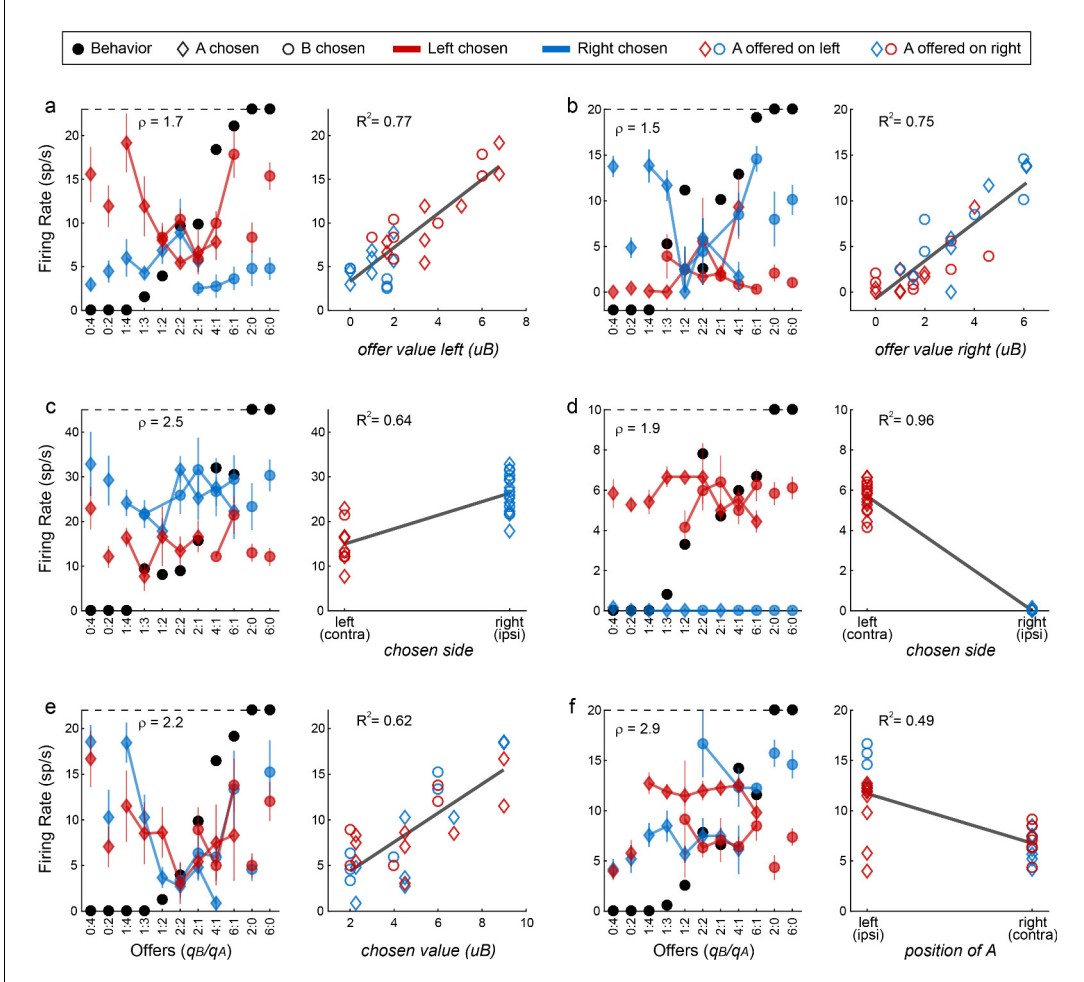

**Figure 9.** Encoding of decision variables, example neurons. (**a**) Response encoding the *offer value left* (from left hemisphere, post-offer time window). In the left panel, firing rates are plotted against the offer type, and black dots show the behavior. Diamonds and circles indicate trials in which the animal chose juice A and juice B, respectively. Red and blue symbols indicate trials in which the animal chose the offer on the left and right, respectively. Thus red diamonds and blue circles correspond to trials in which juice A was offered on the left; red circles and blue diamonds correspond to trials in which juice A was offered on the right. Error bars indicate SEM. The cell activity increased as a function of the value offered on the left, independently of the juice type and the animal's choice. This point is most clear in the right panel, where the same neuronal response is plotted against the variable *offer value left* (expressed in units of juice B). The black line is from a linear regression, and the $R^2$ is indicated in the figure. (**b**) Response encoding the *offer value right* (right hemisphere, late delay). This neuron was recorded in the right hemisphere. (**c**) Response encoding the *chosen side* (right hemisphere, late delay). In this case, the cell activity was nearly binary. It was high when the animal chose the offer on the right (blue) and low when the animal chose the offer on the left (red), irrespective of the chosen juice (diamonds for A, circles for B). (**d**) Response encoding the *chosen side* (right hemisphere, post-juice). (**e**) Response encoding the *chosen value* (right hemisphere, late delay). This response increased as a function of the value chosen by the animal, independent of the juice type and the spatial contingencies. (**f**) Response encoding the *position of A* (left hemisphere, late delay). The cell activity was nearly binary. It was high when juices A and B were respectively offered on the left and on the right; it was low when the spatial contingencies were reversed. Note that the two outliers data points corresponds to forced choices, where juice B was not offered. Conventions in panels b-f are as in panel a.

Second, we defined numerous variables neurons in LO could conceivably encode. These included variables associated with individual juices (*offer value A*, *offer value B*, *chosen value A*, *chosen value B*, *chosen juice*), variables associated with spatial locations (*offer value ipsi*, *offer value contra*, *chosen value ipsi*, *chosen value contra*, *chosen side*), the variable *position of A* capturing the spatial configuration of the offers, and the variable *chosen value*. Of note, variables associated with spatial locations may be defined in Euclidean space (e.g., *offer value left*, *offer value right*) or in relation to the recording hemisphere (e.g., *offer value ipsi*, *offer value contra*). If the internal representation was in Euclidean space, cells encoding the *offer value left* and cells encoding the *offer value right* should

**Table 1.** Results of ANOVAs.

The table reports the results of two ANOVAs. Each column represents one factor, each row represents one time window, and numbers represent the number of cells significantly modulated by the corresponding factor (p<0.001). The bottom row indicates the number of cells that pass the criterion in at least one of the five time windows. The three left-most columns report the results of a 3-way ANOVA. Notably, many cells were modulated by each of the three factors. The right-most column reports the results of a 1-way ANOVA (factor trial type). In total, 301/717 (42%) cells passed the p<0.001 criterion in at least one time window. Neuronal responses that passed this test (N = 565) were identified as task-related and included in subsequent analyses.

| | 3-way | | | 1-way |
|---|---|---|---|---|
| | Offer type | Position of A | Chosen side | Trial type |
| Pre-offer | 1 | 4 | 6 | 8 |
| Post-offer | 55 | 35 | 102 | 121 |
| Late delay | 87 | 40 | 123 | 158 |
| Pre-juice | 39 | 27 | 127 | 121 |
| Post-juice | 58 | 28 | 150 | 155 |
| At least 1 | 154 | 85 | 308 | 301 |

be found in roughly equal proportions in each hemisphere. In contrast, preliminary observations revealed that offer value responses most often encoded the value presented on the ipsi-lateral side. Thus spatial variables included in the analysis were defined in relation to the recording hemisphere (*Table 2*).

Each neuronal response was separately regressed against each variable. Variables that provided a significantly non-zero slope (p<0.05) were said to 'explain' the response. We then generated two population plots. *Figure 10a* illustrates for each time window the number of responses explained by each variable. Since responses could be explained by more than one variable, each response may contribute to multiple bins in this plot. For each response, we also identified the variable that provided the best explanation (highest $R^2$). *Figure 10b* illustrates the population results for this analysis. In this case, each response contributes at most to one bin.

Inspection of *Figure 10b* reveals that few variables were most effective in explaining neuronal responses. To identify a small number of variables that best accounted for the whole data set, we used a stepwise procedure and a best-subset procedure. In the stepwise procedure, we imposed that the marginal explanatory power of each selected variable be at least 5% (see

**Table 2.** Variables defined in the analysis of neuronal responses.

| | Variable | Definition |
|---|---|---|
| 1 | *position of A* | Binary; one if juices A/B are offered on ipsi/contra sides, 0 otherwise |
| 2 | *offer value A* | Value of juice A offered |
| 3 | *offer value B* | Value of juice B offered |
| 4 | *offer value ipsi* | Value offered on the ipsi-lateral side |
| 5 | *offer value contra* | Value offered on the contra-lateral side |
| 6 | *chosen value* | Value of the chosen juice |
| 7 | *chosen value A* | Chosen value if juice A chosen, 0 otherwise |
| 8 | *chosen value B* | Chosen value if juice B chosen, 0 otherwise |
| 9 | *chosen value ipsi* | Chosen value if ipsi side is chosen, 0 otherwise |
| 10 | *chosen value contra* | Chosen value if contra side is chosen, 0 otherwise |
| 11 | *chosen juice* | Binary; one if juice A is chosen, 0 if juice B is chosen |
| 12 | *chosen side* | Binary; one if ipsi side is chosen, 0 if contra side is chosen |

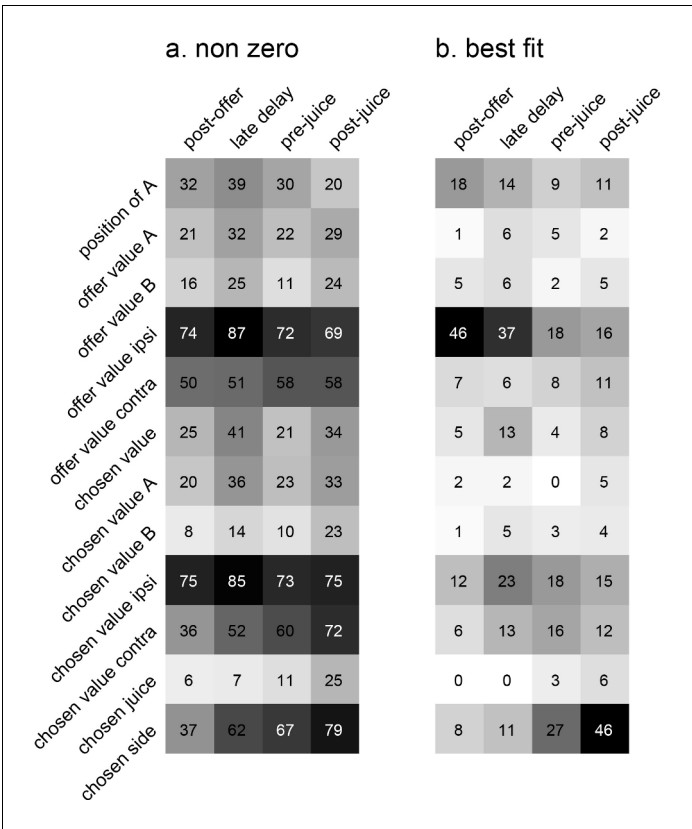

**Figure 10.** Encoding of decision variables, population analysis. (a) Explained responses. This panel shows, for each variable and for each time window, the number of responses explained. For example, 32 responses in the post-offer time window were explained by the variable *position of A*. Because each response could be explained by more than one variable, each response may contribute to multiple bins in this plot. Gray shades recapitulate the cell counts. (b) Best fit. Each response was assigned to the variable that provided the highest $R^2$, and the number of responses obtained are shown here. In this panel, each response contributes to (at most) one bin. Notably, many responses were best explained by variables *offer value ipsi* and *chosen side*.

Materials and methods). In the first four iterations, the procedure selected variables *offer value ipsi*, *chosen side*, *position of A* and *chosen value*, and variables selected in subsequent iterations did not meet the 5% criterion (*Figure 11a*). Together, these four variables explained 457 responses, corresponding to 91% of the responses collectively explained by the 12 variables (*Figure 11b*). The best-subset procedure confirmed this result, showing that the selected variables formed the best possible subset of four variables (*Figure 11c*). We concluded that neurons in LO encode variables *offer value ipsi*, *chosen side*, *position of A* and *chosen value*.

## Discussion

We presented three main results. First, we developed a mouse model of economic choice behavior. The task was very similar to that used in monkey studies, as animals chose between different juices offered in variable amounts. Choice patterns were comparable to those measured for monkeys, although choice variability was generally higher. Of note, initially naïve animals were able to learn the choice task within a few weeks. Second, we showed that economic decisions in mice depend on area LO. Specifically, optogenetic inactivation of LO induced erratic changes of relative value and consistently increased choice variability. Third, we showed that neurons in LO encode different variables reflecting the input (offer values) and output (choice outcome, chosen value) of the decision process. This neural representation closely resembles that previously identified in primates, except that the reference frame in the mouse LO was spatial while that in the monkey OFC was good-based. We next elaborate on each of these results.

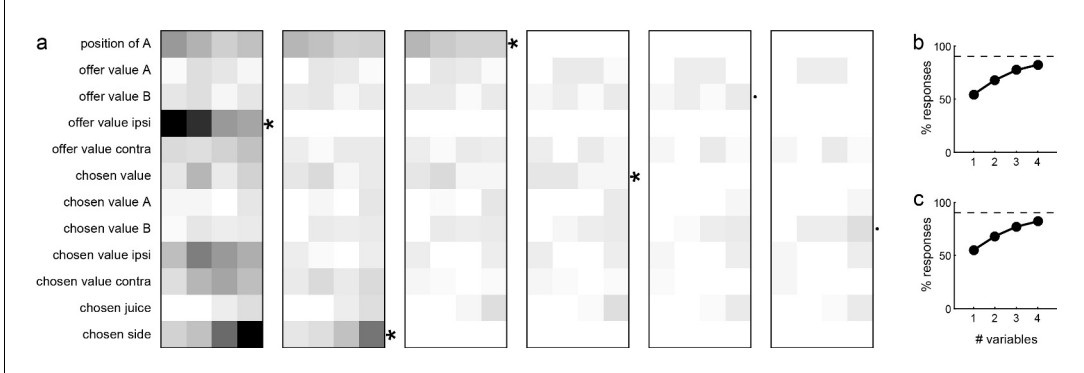

**Figure 11.** Variable selection analysis. (a) Stepwise selection. The leftmost panel is as in *Figure 10b*. At each iteration, we identified the bin with the highest number of responses and we selected the corresponding variable. If the marginal explanatory power was ≥5%, we retained the variable and removed from the pool all the responses explained by that variable. At each iteration, we also verified that the marginal explanatory power of variables selected in previous iterations remained ≥5% once the new variable was selected. In the first four iterations, the procedure selected variables *offer value ipsi*, *chosen side*, *position of A* and *chosen value*. Variables selected in subsequent iterations failed the 5% criterion and were thus rejected. (b) Percent of explained responses, stepwise procedure. The plot illustrates the percent of responses explained (y-axis) as a function of the number of variables (x-axis). On the y-axis, 100% corresponds to the number of responses passing the ANOVA criterion in one of the four time windows (n = 555); the dotted line corresponds to the number of responses collectively explained by the 12 variables included in the analysis (n = 501). The four selected variables explained 457 responses, corresponding to 91% of the responses explained by all 12 variables. (c) Best-subset procedure. The plot illustrates the percent of responses explained (y-axis) as a function of the number of variables (x-axis).

## LO is necessary for economic decisions

Previous work found that lesion or inactivation of orbital cortex in primates and rodents disrupts performance under reinforcer devaluation. This observation is often interpreted as relevant to neuroeconomics, under the assumption that values driving goal-directed behavior are equivalent to values driving economic decisions (*O'Doherty, 2014*; *Padoa-Schioppa and Schoenbaum, 2015*). Our results demonstrate more directly that economic decisions critically depend on the orbital cortex.

Our choice task relied on olfactory stimuli and licking responses. Since viral injections extended beyond LO, one concern was whether choice deficits induced by optical stimulation were due to inactivation of the olfactory bulb or other olfactory areas. Along similar lines, choice deficits could in principle reflect some generic motor impairment. However, insofar as optical stimulation reduced licking, this effect did not explain any difference in choices. Moreover, a LASSO regression analysis attributed optically induced choice deficits to area LO. As a caveat, since the LASSO regression forces most coefficients to zero, one cannot rule out that choice deficits were partly due to inactivation of olfactory regions. A more subtle question is whether LO inactivation disrupted decisions per se, as opposed to olfactory processes to which LO could in principle contribute. We cannot definitively exclude the latter hypothesis based on our current data. However, recent studies indicate that LO inactivation preserves performance in tasks that rely on olfaction but do not require any economic decision (*Wu et al., 2019*), while it affects economic decisions relying on non-olfactory stimuli (*Gardner et al., 2019*; *Gore et al., 2019*). Thus, most likely, LO inactivation in our experiments disrupted economic decisions, not olfactory processes.

Our results on the effects of LO inactivation stand in contrast to those of a recent study that failed to disrupt economic decisions through optogenetic inactivation of area LO in rats (*Gardner et al., 2017*). The discordance is striking because the choice task used in the other study is very similar to ours. Furthermore, as a positive control, the other study reported that LO inactivation affected performance under reinforcer devaluation. Several considerations are in order. In itself, their failure to disrupt economic decisions is not particularly informative. Viral infection is typically more reliable in mice than rats (*Witten et al., 2011*). Furthermore, inactivation through ChR2 stimulation of interneurons is often more effective than inactivation through halo-rhodopsin (*Raimondo et al., 2012*; *Wiegert et al., 2017*), which they used. Thus the most cogent questions pertain to their positive control. In this respect, two elements seem most relevant. First, their experiments were not temporally counterbalanced. All their rats were initially trained in the economic choice task, and tested

under LO inactivation. Subsequently, animals were trained in the reinforcer devaluation task, and tested under LO inactivation. Viral injections were performed after the initial training. Since viral infection takes time (*Witten et al., 2011*), LO inactivation was almost certainly more effective in reinforcer devaluation experiments than in economic choice experiments. Second, a close observation of their behavioral data reveals that in the last three training sessions, the performance of the experimental group was higher than that of the control group (their Figure 6a). If data from these three sessions were combined, the difference between the two groups of rats would presumably be comparable to the difference observed under LO inactivation (their Figure 6c). These various factors, possibly in addition to subtle differences in task design (*Gardner et al., 2017*), can explain the discrepancy between the two studies.

## Representation of subjective values in LO

Our recordings revealed that neurons in LO encoded different variables intimately related to the decision process. Specifically, neurons encoded the spatial configuration of the offers (*position of A*), the decision input (*offer value ipsi*), and the decision output (*chosen side*, *chosen value*). Comparing our results to those of monkey studies, there are notable similarities and interesting differences. On the one hand, three variables represented in LO (*offer value ipsi*, *chosen side* and *chosen value*) are analogous or identical to those previously identified in the primate OFC (*offer value*, *chosen juice* and *chosen value*). On the other hand, a major difference is that input and output in LO are represented in a spatial frame of reference. In contrast, in the primate OFC, input and output are defined in a non-spatial, good-based reference frame (*Padoa-Schioppa, 2011*).

The finding that neurons in LO represent options and values in a spatial reference frame confirms previous reports (*Feierstein et al., 2006*; *Roesch et al., 2006*). However, a striking aspect of our results is that LO cells in our experiments represented values offered on the ipsi-lateral side. The majority of cortical representations are entirely or preferentially contra-lateral. One exception is the olfactory system, as sensory neurons in the olfactory epithelium send their axonal projections ipsi-laterally to the olfactory bulb (*de Olmos et al., 1978*; *Royet and Plailly, 2004*). Offers in our task were represented by olfactory stimuli. Thus one possibility is that the ipsi-laterality found in LO was 'inherited' from the spatial representation in the olfactory bulb. Supporting this hypothesis, connections between the olfactory bulb and piriform cortex, and between piriform and orbital cortices are primarily ipsi-lateral (*Chen et al., 2014*; *Datiche and Cattarelli, 1996*; *Davis and Macrides, 1981*; *Hintiryan et al., 2012*; *Illig, 2005*). Alternatively, ipsi-laterality in LO might be 'endogenous'. In other words, the encoding of offer value might be ipsi-lateral even if offers are represented by visual stimuli. Future research should examine this intriguing issue.

Two groups of responses identified in LO encoded variables *offer value ipsi* and *chosen value*. Importantly, these variables reflect the subjective values of the options and integrate the two dimensions varied in our experiments, namely juice type and quantity. Our result matches a large body of work in human and non-human primates. It also confirms and extends previous observations in rodents (*Gremel and Costa, 2013*; *Hirokawa et al., 2019*; *Roitman and Roitman, 2010*; *Sul et al., 2010*; *van Duuren et al., 2009*; *Zhou et al., 2019*). Interestingly, one earlier study in rats failed to find any systematic relation between the effects induced by changes in reward quantity and those induced by changes in delay (*Roesch et al., 2006*). The reasons for their negative result are not clear, but several factors might contribute. First, in the earlier study, most of the analysis focused on the time window following reward delivery. However, we found that value-encoding responses are most prominent immediately after the offer, and that neuronal responses at juice delivery mostly represent the binary choice outcome. Second, in the earlier study, reduced activity at juice delivery in long delay trials might be due to the animal's inability to predict the reward timing (because of the long delay), as opposed to temporal discounting per se. In fact, at the time of delivery, the subjective value of the juice should no longer depend on the preceding delay. Third, in their study, the two dimensions were never manipulated at the same time. In fact, trials were blocked, both dimensions were fixed within a block, and firing rates were compared across blocks. If the value representation in LO is range adapting as is the value representation in the primate OFC (*Cox and Kable, 2014*; *Padoa-Schioppa, 2009*), neurons in the blocked design might appear untuned. Thus range adaptation might explain why varying the reward quantity had only a modest effect on neuronal activity. In conclusion, future work should re-examine the representation of temporally discounted values in area LO with more suitable task design and data analysis.

### Mechanisms of economic decisions in mice

As noted above, the variables represented in the mouse LO closely resembled those encoded in the primate OFC, except for the fact that they were defined in a spatial reference frame. Modeling work has shown that the three groups of cells identified in the primate OFC are computationally sufficient to generate binary choices, suggesting that economic decisions are formed in a neural circuit within this area (*Padoa-Schioppa and Conen, 2017*). Importantly, current models may be formulated equally well in spatial terms. In other words, neurons encoding the variables identified here are also sufficient to generate binary choices, indicating that economic decisions in mice could be formed within area LO. With this premise, one aspect of our results is noteworthy.

In the primate OFC, neurons encoding different variables, including *offer value A* and *offer value B*, are found in close proximity of each other. This salt-and-pepper distribution suggests that the competition between values happens throughout the OFC and in both hemispheres. In contrast, in the mouse LO, neurons in each hemisphere predominantly represent the value of the good offered on the ipsi-lateral side. This lateralized distribution seems to imply that decisions in our task ultimately involve a competition between the two hemispheres, as previously observed in other domains (*Asanuma and Okuda, 1962*; *Bloom and Hynd, 2005*; *Cazzoli et al., 2009*; *Ferbert et al., 1992*; *Forss et al., 1999*; *Hilgetag et al., 2001*; *Palmer et al., 2012*; *van der Knaap and van der Ham, 2011*). Future work should test this hypothesis more directly, and assess whether the same lateralized organization holds when binary decisions are made in a spatially richer environment (e.g., with free moving animals).

In conclusion, we established a genetically tractable animal model of economic choice behavior, and we demonstrated a clear homology between the mouse LO and the primate central OFC. We gathered strong evidence that economic decisions depend on LO. We also showed that neurons in LO represent the input and the output of the choice process, suggesting that decisions emerge from a neural circuit within this area. With respect to the decision mechanisms, the main difference between mice and primates appears to be that decision variables in LO are represented in a spatial reference frame. The fact that neurons in each hemisphere predominantly represent the value offered on one side suggests that economic decisions in mice involve a competition between the two hemispheres.

## Materials and methods

### Animals and surgical procedures

This study reports on 19 mice of different strains, including C57BL/6J (B6; N = 2; Jackson Laboratory, stock #000664), *Pvalb*$^{tm1(cre)Arbr}$/J (PV-Cre; N = 7; Jackson Laboratory, stock #008069) and Tg (Slc32a1-COP4*H134R/EYFP)8Gfng/J (VGAT-ChR2; N = 10; Jackson Laboratory, stock #014548). Both male and female animals were used for neuronal recordings and optogenetic inactivation of area LO. All mice were >10 weeks old at the time of the experiments. Animals were housed individually and the experiments were conducted in the dark phase of a 12 hr light/dark cycle. Mice were under water restriction. On testing days, they had access to water or sucrose water only during the experiments. All experimental procedures conformed to the NIH *Guide for the Care and Use of Laboratory Animals* and were approved by the Institutional Animal Care and Use Committee (IACUC) at Washington University in St Louis.

Neuronal recordings were conducted with N = 8 VGAT-ChR2 mice. The optogenetic inactivation experiments were conducted on N = 2 VGAT-ChR2 mice and on N = 5 PV-Cre mice with AAV-DIO-ChR2 injections. The control group included N = 2 B6 mice (no injection) and N = 1 PV-Cre mouse injected with saline. The same cannulas were implanted in non-injected and in injected mice, and surgery damage was very similar in the two groups. Mice were randomly assigned to different groups. For the neuronal recordings, we initially set up to collect a dataset comparable to those we typically record in monkeys, consisting of 500–1000 cells. For the inactivation experiments, we did not perform any ex-ante power analysis. After the pilot experiments (N = 2 mice, VGAT), we decided to replicate the experiments in 5 mice and 10–15 sessions per animal.

All surgeries were conducted under general anesthesia induced with Isoflurane, alone or in combination with Ketamine. A titanium head plate implanted on the bregma was used to restrain the animal and as a landmark for the neuronal recordings. The craniotomy, slightly larger than the target

recording area, typically spanned 2.0–3.5 mm anterior to the bregma and 0.5–1.8 mm lateral to the midline. For the optogenetic experiments, we implanted cannulas with an optical fiber (200 μm core diameter) bilaterally at 2.7 mm anterior, 1.4 mm lateral. The tip of the cannula was placed at 1.1 mm ventral to the brain surface. In relevant experiments, adeno-associated virus (AAV.EF1a.DIO.hChR2 (H134R)-eYFP.WPRE.hGH; Addgene 20298; titer = $7*10^{12}$ vg/ml) or saline was injected with a Hamilton syringe at 2.7 mm anterior, 1.4 mm lateral, 1.8 mm ventral from the brain surface. Specifically, we injected 100 nl over 10 min, and we waited 2 min before retracting the syringe.

## Economic choice task

We designed the choice task to resemble as much as possible the task used for monkeys (*Padoa-Schioppa and Assad, 2006*). During the experiment, the mouse was placed in a plastic tube with the head fixed. Two odor delivery systems and two liquid spouts were placed symmetrically on the left and on the right of the animal head, and close to the mouth. In each session, the animal chose between two liquid rewards offered in variable amounts, delivered from left and right lick ports. We refer to the liquid rewards as 'juices' for uniformity of language with the monkey studies. On any trial, the offered juice types were signaled with different odors, and juice amounts were signaled by the corresponding odor concentration. Before each trial, a vacuum sweep removed the odors remaining from the previous trial. Immediately thereafter, the two odors (the offers) were presented simultaneously from two directions (left, right). The odor presentation lasted for 2.8 s, after which the animal indicated its choice by licking one of the two spouts. The liquid spouts were fixed and did not retract during odor presentation. The response period started with an auditory 'go' cue and licking before the go cue was disregarded. Licking was detected by two photodiodes located posterior to the liquid spouts. If the animal did not respond within 4 s, the trial was aborted (*Figure 1*). In forced choices, where only one juice was offered, trials in which the animal licked the wrong spout were considered errors. Such cases were almost inexistent in monkey experiments, but occurred sometimes in mouse experiments (*Figure 1*). Upon error trials, we aborted the trial and repeated the same offer in the subsequent trial.

Throughout the experiments, we used the same two liquid rewards, namely water and 12% sucrose water. We always labeled sucrose water as 'juice A' and water as 'juice B'. In the vast majority of sessions, sucrose water was preferred to water. This point can be observed in *Figure 2a*, showing that in 96% of sessions we measured $\rho > 1$. Odors were mint and 4-Pentenoic acid. The association between the juices and the odors varied pseudo-randomly across mice. The juice amount offered to the animal varied between 0 and 6 drops (eight drops for mice #39 and #53). The amount was monotonically related to the represented odor concentration, which varied roughly linearly on a log2 scale (e.g., odor levels 1, 2, 4 and 8 ppm representing quantities 1, 2, 3 and 4 of juice). In each session, the left/right location of the two juices varied pseudo-randomly from trial to trial. Juice delivery took 150–900 ms depending on the juice amount (~150 ms per quantum). Independent of the amount, the reward period was kept constant and equal to 900 ms, to maintain a consistent trial duration. Excepting this rule, when mice #39 and #53 received 8 drops of juice, the reward period lasted 1.2 s. Odor presentation for the next trial started 650 ms after the end of the reward period. For most mice, we used higher odor concentrations to represent larger amounts of juice. However, we also trained two mice in a 'flipped' version of the task, in which higher odor concentrations represented smaller amounts of juice. Although training took longer, the two mice reached a similar level of performance in the choice task (*Figure 2*).

Typically, mice performed the task for ~30 min each day, during which they completed ~300 correct trials and received 0.8–1.6 ml of liquid reward. The multiplicity of offer types was fixed within each session and the offer type was randomly selected at the beginning of each trial. Sessions typically included ~40% of forced choices (20% for each juice). In forced choice trials, error trials were followed by a 3 s additional delay (with white noise). (For some sessions in mice #36 and #41, the delay lasted 5 s; in mouse #39, the delay lasted 1 s.)

## Training protocol

With experience, our training protocol became more standard. Eventually, training developed in four steps. (1) Mice were trained in a direction discrimination task. We presented the odor from the left or from the right in random alternation, and we delivered the juice only when the animal licked

the corresponding spout. Mice typically took ~10 days to reach 80% accuracy. (2) We introduced the association between odor concentration and juice quantity. We used the same scheme as in (1), but the odor was presented at different concentrations and coupled with different juice quantities. Mice typically took ~3 days to reach 80% accuracy. (3) We introduced the association between different odors and different juice types. Specifically, we repeated steps (1) and (2) using a second odor and a second juice type (in some cases, we experimented with 3 or four juice types). Mice typically took 6–10 days to reach 80% accuracy. (4) We presented mice with the full choice task, where animals choose between the two juice types offered in variable amounts (*Figure 1*). We trained the animal on the choice task for at least 5 days before starting the experiments.

Of the two mice trained in the 'flipped' version of the task, one (#39) was naïve while the other (#53) had already been trained in the standard task. Both mice took ~20 extra days to reach performance level.

## Optical stimulation, neuronal recordings, and histology

Optical stimulation was performed with blue light (473 nm Blue DPSS Laser, Shanghai Laser). Core fibers (200 μm, Doric) were connected through two cannulas inserted bilaterally in area LO. To precisely control the stimulation timing, we used an acousto-optic device (AO modulator/shifter, Optoelectronics) and an associated RF Driver MODA110-B4-30 (Optoelectronics). To inactivate area LO, we typically used 3–9 mW intensity, 10 ms pulses and 10–33 Hz frequency. The stimulation started at the beginning of odor presentation and lasted 3.8 s (i.e., throughout the offer period plus 1 s). In most sessions, stimulation conditions (OFF or ON) varied pseudo-randomly on a trial-by-trial basis. In a subset of sessions, trials were divided in blocks of 20–30 trials. The optical inactivation experiments were conducted on N = 2 VGAT-ChR2 mice (29 sessions total), N = 5 PV-Cre mice injected with AAV-DIO-ChR2 in area LO (78 sessions total), and N = 3 control mice (39 sessions total).

We recorded the spiking activity of individual neurons from LO of eight mice. Recording locations spanned 2.5–3.1 mm anterior, 1.0–1.7 mm lateral and 1.0–2.0 mm ventral. Extracellular activity was recorded with a 16 channel array, with or without optical fiber (Neuronexus). The array was advanced before each session. Electric signals were amplified (10,000 gain) and band-pass filtered (300 Hz - 6 kHz; Neuralynx). Spikes were identified with a threshold, digitized (40 kHz; 1401, Cambridge Electronic Design), and stored to disk for off-line spike sorting (Spike2, Cambridge Electronic Design).

At the end of the recording experiments, we injected a dye (DiI) roughly at the center of the recording region. The animal was then perfused. The brain was extracted, mounted on an optimal cutting temperature compound and frozen. Subsequently, the brain was sliced (approximately 33 μm sections) with a low-temperature Cryostat (Leica Biosystems) and pasted on cover glass. Sections were then examined and photographed under a fluorescence microscope (Leica DMI6000 B microscopy). *Figure 12* illustrates the reconstructed locations of recordings.

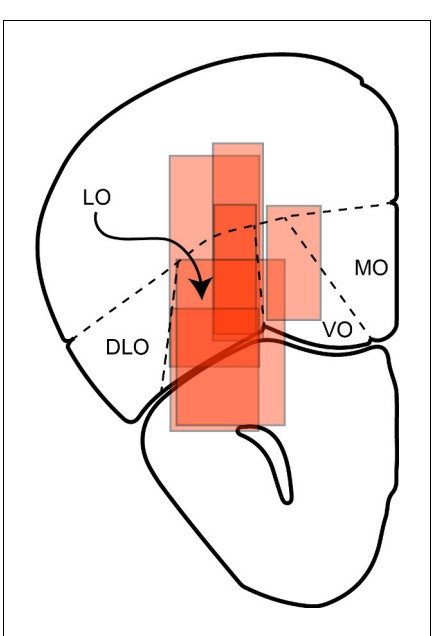

**Figure 12.** Reconstructed locations of neuronal recordings for six mice. Recording regions for all animals were transferred on the same hemisphere. Approximate AP coordinate of this section is bregma 2.68 mm, interaural 6.48 mm.

## Data analysis, behavior

All the analyses of behavioral and neuronal data were conducted in Matlab (MathWorks). Behavioral choice patterns were analyzed using logistic regression. The basic logistic model was written as follows:

$$choiceB = 1/(1 + \exp(-X))$$
$$X = a_0 + a_1 \log(q_B/q_A) \qquad (1)$$

where *choice B* equals one if the animal chose

juice B and 0 otherwise, $q_A$ and $q_B$ are the quantities of juices A and B offered to the animal, and A is the preferred juice. Forced choice trials were excluded. The logistic regression provided an estimate for parameters $a_0$ and $a_1$. By construction, $a_0 < 0$ and $a_1 > 0$. The relative value of the two juices was defined as $\rho = \exp(-a_0/a_1)$. In essence, in any given session, $\rho$ was the amount of juice B that, if offered against 1A, made the animal indifferent between the two juices. The sigmoid steepness was defined as $\eta = a_1$. The steepness (also termed inverse temperature) is inversely related to choice variability.

In further analyses, we examined the possible effects on choices of three additional factors. First, the side bias was examined with the following model:

$$choice\,B = 1/(1+\exp(-X))$$
$$X = a_0 + a_1 \log(q_B/q_A) + a_2(\delta_{A,\,right} - \delta_{B,\,right}) \tag{2}$$

where $\delta_{J,\,right}$ = 1 if juice J was offered on the right and 0 otherwise, and J = A, B. This logistic fit returned two sigmoid functions with the same steepness and different flex points. The side bias was quantified as $\varepsilon = -a_2/a_1$. A measure of $\varepsilon > 0$ indicated that, other things equal, the animal tended to choose the juice offered on the right.

Second, choice hysteresis (*Padoa-Schioppa, 2013*) was examined with the following model:

$$choice\,B = 1/(1+\exp(-X))$$
$$X = a_0 + a_1 \log(q_B/q_A) + a_2(\delta_{n-1,\,A} - \delta_{n-1,\,B}) \tag{3}$$

where $\delta_{n-1,\,X}$ = 1 if in the previous trial the animal chose juice X and 0 otherwise, and X = A, B. Choice hysteresis was quantified as $\xi = -a_2/a_1$. A measure of $\xi > 0$ indicated that, other things equal, the animal tended to choose the same juice chosen in the previous trial.

Third, direction hysteresis was examined with the following model:

$$choice\,B = 1/(1+\exp(-X))$$
$$X = a_0 + a_1 \log(q_B/q_A) + a_2(\delta_{n-1,\,posA} - \delta_{n-1,\,posB}) \tag{4}$$

where $\delta_{n-1,\,pos\,B}$ = 1 if juice B was offered in the same spatial position as that chosen in the previous trial and 0 otherwise, and $\delta_{n-1,\,pos\,A} = 1 - \delta_{n-1,\,pos\,B}$. Direction hysteresis was quantified as $\theta = -a_2/a_1$. A measure of $\theta > 0$ indicated that, other things equal, the animal tended to choose the juice offered on the same side as the side chosen in the previous trial.

All the logistic models described so far analyzed choices in the absence of optogenetic manipulations. To quantify the effects of these manipulations, we constructed additional logistic models. The basic effects of LO inhibition on the relative value and choice variability were quantified with the following model:

$$choice\,B = 1/(1+\exp(-X))$$
$$X = (a_0 + a_1 \log(q_B/q_A))\delta_{stim,OFF} + (a_2 + a_3 \log(q_B/q_A))\delta_{stim,ON} \tag{5}$$

where $\delta_{stim,\,ON}$ = 1 in stimulation trials and 0 otherwise, and $\delta_{stim,\,OFF} = 1 - \delta_{stim,\,ON}$. In essence, *Equation 5* repeats *Equation 1* twice, once for trials without stimulation and once for trials with stimulation. Hence, the logistic fit returns two sigmoid functions that differ for their flex point and for their steepness. The effects of stimulation were quantified by comparing $\rho_{stim\,OFF} = \exp(-a_0/a_1)$ and $\rho_{stim\,ON} = \exp(-a_2/a_3)$, and by comparing $\eta_{stim\,OFF} = a_1$ and $\eta_{stim\,ON} = a_3$.

To test whether LO inactivation specifically affected the side bias, we constructed the following model:

$$choice\,B = 1/(1+\exp(-X))$$
$$X = (a_0 + a_1 \log(q_B/q_A) + a_2(\delta_{A,right} - \delta_{B,right}))\delta_{stim,OFF} +$$
$$(a_3 + a_4 \log(q_B/q_A) + a_5(\delta_{A,right} - \delta_{B,right}))\delta_{stim,ON} \tag{6}$$

The test was conducted by comparing $\varepsilon_{stim\,OFF} = -a_2/a_1$ and $\varepsilon_{stim\,ON} = -a_5/a_4$. We constructed analogous logistic models to test whether LO inactivation specifically affected choice hysteresis or direction hysteresis.

## Data analysis, neuronal activity

Preliminary observations revealed that the activity of neurons in LO varied as a function of the offered and chosen juices, but also depended on the spatial contingencies of the choice task. This is unlike what we found in the primate OFC, where firing rates are independent of the spatial contingencies (*Grattan and Glimcher, 2014*; *Padoa-Schioppa and Assad, 2006*). Thus the present analysis was designed to capture the spatial components of the choice task. Apart from this aspect, our analyses closely resembled those previously conducted in monkey studies. Neuronal activity was examined in 5 time windows: pre-offer (0.6 s preceding the offer); post-offer (0.2–0.8 s after offer on); late delay (0.6–1.2 s after offer on); pre-juice (0.6 s preceding juice delivery); post-juice (0.6 s following the beginning of juice delivery). An 'offer type' was defined by two juice quantities offered to the mouse, independent of the spatial configuration and the animal's choice. A 'trial type' was defined by an offer type, a spatial configuration (e.g., juice A on the left), and a choice. A 'neuronal response' was defined as the activity of one neuron in one time window as a function of the trial type. Sessions typically included 10–12 offer types (including forced choices), and ~30 trial types. Error trials in forced choices were excluded from the analysis. We also excluded from the analysis trial types with $\leq 2$ trials. Neuronal responses were constructed by averaging spike counts across trials for each trial type.

Our analyses aimed at identifying the variables encoded in area LO and proceeded in steps. First, the activity of each cell in each time window was examined with a 3-way ANOVA (factors offer type ×position of A × chosen side), followed by a 1-way ANOVA (factor trial type). The latter was used to identify 'task-related' responses. Specifically, we imposed a significance threshold of $p < 0.001$, and neuronal responses that passed this criterion were included in subsequent analyses.

Second, we defined a series of variables that neurons in LO might potentially encode (*Table 2*). For each neuronal response, we performed a linear regression against each variable (separately). If the regression slope differed significantly from zero ($p < 0.05$), the variable was said to 'explain' the neuronal response. For each variable, the regression also provided an $R^2$. For variables that did not explain the neuronal response, we arbitrarily set $R^2 = 0$.

Third, we conducted population analyses to identify a small subset of variables that best accounted for our data. Based on the regressions, we identified for each response the variable that provided the best fit (highest $R^2$). We then computed the number of responses best explained by each variable, separately for each time window (*Figure 10*). To identify the variables that best accounted for the whole population, we proceeded as in previous studies, using two methods of variable selection – stepwise and best subset (*Glantz and Slinker, 2001*; *Padoa-Schioppa and Assad, 2006*). The stepwise method is an iterative procedure. In the first step, we selected the variable that provided the highest number of best fits within any time window, and we removed from the data set all the responses explained by this variable. In the second step, we repeated the procedure with the residual data set. We defined the 'marginal explanatory power' of a variable X as the percent of task-related responses explained by X and not explained by any other selected variable. At each step, we imposed that the marginal explanatory power of each selected variable be $\geq 5\%$. We then continued the procedure until additional variables failed to meet the 5% criterion. In contrast, the best subset method is an exhaustive procedure. For this analysis, we pooled responses from different time windows. For each possible subset of $d$ variables, we computed the number of responses explained in the data set, and we identified the subset that explained the highest number of responses. We repeated this procedure for $d$ = 1,2,3...

## Acknowledgements

We thank Grant Black for help with animal training, Andreas Burkhalter for help with the histology, and Alessandro Livi for help with the figures. We also thank Ed Han, Ahmad Jezzini, Lex Kravitz, Alessandro Livi, Weikang Shi and Manning Zhang for comments on previous versions of the manuscript. This work was supported by the National Institutes of Health (grant number R21-DA042882 to CPS).

## Additional information

### Funding

| Funder | Grant reference number | Author |
| --- | --- | --- |
| National Institute on Drug Abuse | R21-DA042882 | Camillo Padoa-Schioppa |

The funders had no role in study design, data collection and interpretation, or the decision to submit the work for publication.

### Author contributions

Masaru Kuwabara, Data curation, Software, Formal analysis, Investigation; Ningdong Kang, Methodology, Design of experimental apparatus; Timothy E Holy, Conceptualization, Resources, Software, Supervision, Project administration; Camillo Padoa-Schioppa, Conceptualization, Software, Formal analysis, Supervision, Funding acquisition, Project administration

### Author ORCIDs

Camillo Padoa-Schioppa (iD) https://orcid.org/0000-0002-7519-8790

### Ethics

Animal experimentation: All experimental procedures conformed to the NIH Guide for the Care and Use of Laboratory Animals and were approved by the Institutional Animal Care and Use Committee (IACUC) at Washington University in St Louis (protocol # 20160167).

### Decision letter and Author response

Decision letter https://doi.org/10.7554/eLife.49669.sa1
Author response https://doi.org/10.7554/eLife.49669.sa2

## Additional files

### Supplementary files

• Transparent reporting form

### Data availability

Data and analysis files included as supplementary information.

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
