## [Decision Letter]

**Acceptance summary:**

This study examined the role of the orbitofrontal cortex (OFC) in value-based decisions in mice. The authors developed an economic decision task in mice that is similar to their primate studies, enabling comparative analysis between the species. The results indicate that OFC plays an important role in this task, in contrast to some previous work in rats. The authors found interesting similarities as well as differences in neuronal responses in the mouse OFC compared to the monkey OFC. Together, these results provide important results that will shape our understanding of the OFC's role in value-based decisions.

**Decision letter after peer review:**

Thank you for submitting your article "Neural mechanisms of economic choices in mice" for consideration by *eLife*. Your article has been reviewed by three peer reviewers, and the evaluation has been overseen by a Reviewing Editor and Christian Büchel as the Senior Editor. The reviewers have opted to remain anonymous.

The reviewers have discussed the reviews with one another and the Reviewing Editor has drafted this decision to help you prepare a revised submission.

Summary:

Specifically, whether the orbitofrontal cortex (OFC) plays a critical role in value-based decisions remains controversial, particularly in rodents. This study established a mouse model to study value-based decisions by adapting to mice the economic decision task that was developed by Padoa-Schioppa and Assad, 2006, and has been a powerful paradigm in primates. The authors show that optogenetic inactivation of the lateral orbitofrontal cortex (LO) impaired the behavioral performance in this task. Furthermore, the authors found that LO neurons in mice showed the types of activity observed in monkeys although LO neurons in mice exhibited biases in activity related to the choice/movement direction.

All the reviewers thought that this is an important study addressing long-standing questions regarding the role of OFC in economic decision making. Furthermore, they thought that establishing a mouse model is a great step toward further mechanistic studies. The manuscript is clearly written, and the data are well presented. The reviewers, however, raised some technical concerns that require further investigations.

Essential revisions:

1) Most importantly, the reviewers thought that it remains unclear whether the optogenetic inactivation is confined to the intended area (LO). From the data provided, it is unclear whether the authors can exclude the possibility that observed behavioral effects (increased choice variability) are due to an off-target effect such as inactivation of the piriform cortex or anterior olfactory nucleus which are right beneath the LO. Due to the small size of mice, compared to rats, this issue appears to be particularly important. This would be an issue for both the GAD mouse but also the PV-Cre mice, as the virus injections do appear to include more ventral structures (Figure 3G, right).

2) Somewhat related to the above issue, we thought that it remains unclear whether the LO inactivation influenced the more basic sensorimotor choice behavior as opposed to specifically influencing economic decisions. It would be very helpful to examine the effect of LO inactivation in a simpler odor-guided two choice behavior.

3) The reviewers ask for more detailed descriptions of methods (reviewer 2, comments 1-3; reviewer 3, comments #2). Please clarify.

Reviewer #1:

The authors conduct a nice study, implementing an economic decision task in mice that is quite similar to their primate studies, enabling comparable analyses between the species. They find that mice exhibit similar choice patterns to primates, albeit with more trial-to-trial variability, in part reflecting more prominent biases. Bilateral optogenetic inactivation of LO, either by ChR2 activation of GAD or PV neurons, increases choice variability and often a reliance on biases; this positive result is in contrast to previous work in rats. Electrophysiological recordings from LO reveal qualitatively similar responses to what the authors have observed in primates. An interesting difference is that value and choice representations in the mouse appear to be spatial in contrast to primates.

Overall, I have no major issues with the analyses performed in this study. My biggest concern, however, is that their optogenetic result reflects off-target effects of the piriform cortex or accessory olfactory bulb, which are just ventral to LO. This could be particularly problematic as their task is olfactory based, and their optogenetic result (increased choice variability) could in principle reflect reduced ability to discriminate the olfactory cues. Additionally, their results differ from recent work in the rat (Gardner et al., 2017), which they discuss. However, it is worth noting that the rat brain is much bigger with a cortex that is twice as thick as the mouse. It could be that, because of the smaller distance between LO and the AOB/piriform in the mouse, light from a fiber placed in LO can more easily reach these deeper structures in the mouse than the rat. This would be an issue for both the GAD mouse but also the PV-Cre mice, as the virus injections do appear to include more ventral structures (Figure 3G, right). Have the authors confirmed that olfactory detection is not impaired with their optogenetic inactivations? Do they have any catch trials that they could analyze to this effect? This is a fundamental issue that needs to be clarified.

Can the authors further confirm that there were not off target effects of overlying motor cortex, which was also infected with virus (Figure 3G, left)? Were they any changes in response times, number of licks, etc.?

Additionally, in the Discussion section, when the authors attempt to reconcile their results with the Gardner paper, they include the following statements: "Second, a close observation of their behavioral data reveals that in the last 3 training sessions, the performance of the experimental group was higher than that of the control group (their Supplementary Figure 2A). If data from these three sessions were combined, the difference between the two groups of rats would presumably be comparable to the difference observed under LO inactivation (their Supplementary Figure 2C)." Out of curiosity I attempted to track this down, but I'm confused. Are they referencing the Gardner et al., 2017 paper? Supplementary Figure 2 in that paper does not have A or C subpanels, and I couldn't make sense of their statement from that figure. Please clarify.

Reviewer #2:

In their manuscript, "Neural mechanisms of economic choices in mice", the authors examine the role lateral orbitofrontal cortex (LO) in economic choice behavior in mice. In the current manuscript, they adapt the choice task used in non-human primates for mice to test whether LO is necessary for choice behavior and examine how variables that affect choices are encoded in single neuron responses. The authors demonstrate a role for LO in controlling choice behavior and show similar patterns of neural responses as observed in monkeys. However, the neural responses related to preference are more affected by spatial aspects of the task in mice. The ability to carry out these kinds of studies in a mouse model opens the door for tests of mechanism not possible in the non-human primate model. There are a few areas that the authors could clarify for a more complete description of their findings:

1) For readers familiar with the previous work in primates, the description of the rodent task would benefit from greater clarification of details. Describing the liquid reward as 'juices' creates an expectation that mice are choosing between apple juice, grape juice, fruit punch, tea, etc. The Materials and methods describe training mice to choose between different quantities of water and 12% sucrose solution. In the rodent literature, these are not typically referred to as juices, but just fluid reward. Here juices A and B are used to refer to the preferred and non-preferred fluid – in two-bottle preference tests, mice typically prefer sucrose. In this paradigm, how often is water the preferred option? As mice are fluid restricted and sucrose solution presents an osmotic challenge, does preference change over the course of a session, or over the course of days of testing?

2) There does not seem to be an introduction of a wider range of juice offers or olfactory cues. Indeed, it would be very surprising if mice (or rats) could flexibly adapt and produce reliable behavior in the more complex version of the task that monkeys perform. I would suggest making it more explicitly clear in the Results that the rodent menu is reduced compared to the monkey, and include in the Discussion the experimental advantages to the rodent model for elucidating mechanism as well as the reduced flexibility in the behaviors available to study.

3) Were the spouts continuously available, or did they retract between trials? If spouts are continuously available, behavior between trials may provide another index of spatial biases in the animals.

4) Licks before the 'go' cue were discarded. Did this 'anticipatory' licking correspond with the spatial encoding of LO neurons?

5) In the optogenetic stimulation experiments, the timing of stimulation was set to coincide with odor cue onset. Related to #4, if a mouse were engaged in anticipatory licking, did optogenetic stimulation affect that behavior as well as choice? Was stimulation ever delivered in a different temporal window?

Reviewer #3:

In the manuscript "Neural mechanisms of economic choices in mice", the authors developed a mouse behavior paradigm to examine economic decisions based on similar behavioral tasks in non-human primates. The authors examined the involvement of the lateral OFC in this task using optogenetic silencing methods and performed extracellular recording from individual neurons in the lateral OFC to investigate neuronal coding of various variables relevant to the economic decision process. The authors found that inactivation of LO disrupted mice's performance of the economic decision task and increased the stereotypy of choices independent of subjective values. The authors also found that neuronal responses in different trial epochs encoding different aspects of the economic choices, with early responses mostly encoding offer value on the ipsilateral side, late responses mostly encoding the spatial direction of choices, and some other neurons encoding the chosen value on the ipsilateral side. This is a timely study that successfully translated economic decision paradigm from primates to rodents and provided sophisticated analysis of the neuronal activity in the lateral OFC to show that OFC neurons encode important behavioral variables related to economic decisions. The compelling evidence provided in this study support that the rodent OFC plays a similar role in economic choice behavior as in primates, which opens up the possibility to investigate at more mechanistic level the neural circuit basis of economic decision-making.

1) While successfully training mice to perform this advanced economic decision task is certainly a prize, this behavioral task is quite complex that contains multiple cognitive components and requires several stages of training. This leads to my concern of whether silencing LO specifically impaired economic decision-making, or it might have also disrupted other processes involved in this task. For example, would LO inactivation also disrupts the sensorimotor process, which can be critical for the choice behavior based on odor-reward association? From the data plot in Figure 3, it seems that LO inactivation also disrupted the performance in the easiest trials, where there was only one odor and one rewarded side, and correct responses may only require the odor-reward association. A potential experiment to test this possibility is to inactivate LO in a simpler odor discrimination 2AFC task. If mice's performance is also impaired, then it would be difficult to tell whether LO was only involved in economic decisions or it is generally required for sensorimotor choice behavior. In fact, given the broad projection of OFC neurons and the nonselective nature of silencing methods, it may not be unexpected that the inactivated OFC neurons may also be involved in more general choice processes. If this is the case, the authors may need to confine their interpretation of inactivation results and rely more on the analysis of the recording data to understand the function of OFC neurons.

2) The authors did not provide sufficient details of virus injection, such as the volume and titer of injected virus, and the injection methods etc, which can influence the range of expression. Although the author optically targeted the LO region, the virus infection regions seem to be quite large (Figure 1G). Photo stimulation may potentially influence neighboring regions. While this issue can be difficult to address experimentally given the difficulty of the task training, the authors should at least discuss the potential caveats.

3) The authors found that neuronal activity in different time epochs encode different aspects of the economic decisions. It would be interesting to perform optogenetic silencing restricted in different trial epochs. This would be an optional set of experiments that might not be required to support the current conclusions in this study, but this would provide more informative results to support the specific role of OFC in the economic decisions.

4) It is still very intriguing why LO mainly encode values associated with the ipsilateral side. The authors speculated that this may be specific to the olfactory sensory pathways. This explanation can be better supported if the authors could provide references or evidence supporting that the circuit pathways linking olfactory bulb eventually to OFC are also primarily ipsilateral.

[Editors' note: further revisions were suggested prior to acceptance, as described below.]

Thank you for submitting your article "Neural mechanisms of economic choices in mice" for consideration by *eLife*. Your article has been reviewed by three peer reviewers, and the evaluation has been overseen by a Reviewing Editor and Christian Büchel as the Senior Editor. The reviewers have opted to remain anonymous.

The reviewers have discussed the reviews with one another and the Reviewing Editor has drafted this decision to help you prepare a revised submission.

Overall, the reviewers thought that the authors have addressed most of the previous concerns by additional analysis and revising the manuscript. However, they thought that the issue of regional specificity of optogenetic inhibition should be discussed in a more balanced way. The reviewers agreed that the new analyses using the lasso regression cannot completely rule out the possibility that some, if not all, of the results can be accounted for by unintended inactivation of underlying olfactory areas. After discussion among the reviewers, we agreed that the authors should discuss this caveat explicitly.

---

## [Author Response]

Essential revisions:1) Most importantly, the reviewers thought that it remains unclear whether the optogenetic inactivation is confined to the intended area (LO). From the data provided, it is unclear whether the authors can exclude the possibility that observed behavioral effects (increased choice variability) are due to an off-target effect such as inactivation of the piriform cortex or anterior olfactory nucleus which are right beneath the LO. Due to the small size of mice, compared to rats, this issue appears to be particularly important. This would be an issue for both the GAD mouse but also the PV-Cre mice, as the virus injections do appear to include more ventral structures (Figure 3G, right).

We agree that this is a critical issue, and we addressed it with a new analysis linking the choice deficit induced by optical stimulation with the infected volume for different brain areas. More specifically, we focused on the 5 PV-Cre mice used for the key inactivation experiments. As noted by the reviewers, viral infections were not confined to area LO. Based on the histology (Figure 3G), we quantified the infected volume for each area surrounding LO. Across mice, we found infected tissue in 12 brain areas, namely FrA, PrL, MO, VO, LO, DLO, OB, M1, M2, CgI, AI and EX.

The primary effect of the optogenetic manipulation on choices was to decrease the sigmoid steepness η (Figure 5). For each session, we quantified this effect by defining the steepness stimulation index (SSI):

SSI = 2 * (η_stim ON_ – η_stim OFF_) / (η_stim ON_ + η_stim OFF_)

For each mouse, we defined the choice deficit as CD = –mean(SSI), where the average was across sessions and the minus sign made it so that more severe behavioral impairments corresponded to larger CDs.

To address the issue at hand, we conducted a regression analysis in which we sought to explain the choice deficit (CD) as a function of the infected volume in each area. Since the number of infected brain areas exceeded the number of animals, we used a LASSO regression. This procedure limits the number of covariates. Furthermore, if inactivation of a particular area reduced choice performance, the regression coefficient obtained for that area should be positive. Of all the areas included in the analysis, only 3 presented a non-zero coefficient, and only one – namely LO – presented a positive sign. In other words, only for area LO did the volume of infection correlate with the behavioral choice deficit.

This result strongly supported our interpretation and is described in a new section in the revised manuscript (subsection “*Choice deficits reflect LO inactivation”*, Figure 6, Figure 7).

2) Somewhat related to the above issue, we thought that it remains unclear whether the LO inactivation influenced the more basic sensorimotor choice behavior as opposed to specifically influencing economic decisions. It would be very helpful to examine the effect of LO inactivation in a simpler odor-guided two choice behavior.

Again, we agree that in principle choice deficits induced by optical stimulation could be caused by some motor impairment as opposed to interference with the economic decision process per se. The results of the LASSO analysis described above, which included areas M1 and M2, argue against this hypothesis. However, to further scrutinize this issue, we examined the licking activity in the 5 PV-Cre mice. We found that optical stimulation did affect licking to some extent. Specifically, in the relevant time window, optical stimulation reduced the average number of licks by roughly 8%. While quantitatively modest, this reduction was statistically significant. Of course, this observation does not imply that choice deficits were caused by motor impairments. The crucial question is whether the choice deficit – i.e., the decrease in sigmoid steepness – was at all correlated with the motor deficits across sessions and subjects. To address this issue, we defined for each session the lick stimulation index (LSI) where nlicks was the average number of licks per trial:

LSI = 2 * (nlicks_stim ON_ – nlicks_stim OFF_) / (nlicks_stim ON_ + nlicks_stim OFF_)

In essence, LSI quantified the effect optical stimulation on licking. We thus examined the relation between LSI and the steepness stimulation index (SSI) defined above (point 1). Critically, we did not find any correlation between the two measures in any of the 5 animals (all r ≤ 0.2; all p ≥ 0.35). On this basis, we concluded that choice deficits were caused by specific interference with the decision process and not by generic motor impairments.

These results are reported in the new section (subsection “*Choice deficits reflect LO inactivation”*, Figure 6, Figure 7).

3) The reviewers ask for more detailed descriptions of methods (reviewer 2, comments 1-3; reviewer 3, comments #2). Please clarify.

We revised the manuscript to address each of these comments, as described below in our response to the individual reviewers.

Reviewer #1:[…] My biggest concern, however, is that their optogenetic result reflects off-target effects of the piriform cortex or accessory olfactory bulb, which are just ventral to LO. This could be particularly problematic as their task is olfactory based, and their optogenetic result (increased choice variability) could in principle reflect reduced ability to discriminate the olfactory cues. Additionally, their results differ from recent work in the rat (Gardner et al., 2017), which they discuss. However, it is worth noting that the rat brain is much bigger with a cortex that is twice as thick as the mouse. It could be that, because of the smaller distance between LO and the AOB/piriform in the mouse, light from a fiber placed in LO can more easily reach these deeper structures in the mouse than the rat. This would be an issue for both the GAD mouse but also the PV-Cre mice, as the virus injections do appear to include more ventral structures (Figure 3G, right). Have the authors confirmed that olfactory detection is not impaired with their optogenetic inactivations? Do they have any catch trials that they could analyze to this effect? This is a fundamental issue that needs to be clarified.

We addressed this issue above (Essential revisions, point 1). In essence, we conducted a LASSO regression analysis that linked choice deficits to viral infection – and thus inactivation – in different brain areas including the olfactory bulb. The only area for which we found a significantly positive association was LO.

Aside from this result, it is worth mentioning two posters presented at the last SfN meeting. In one study (Gore et al., 2019), rats performed a juice choice task very similar to ours, except that offers were represented by visual stimuli. In other words, the design was nearly identical to that of Gardner et al., 2017. Gore and colleagues used optogenetics to inactivate LO and found that this manipulation induced choice deficits very similar to the ones reported here. Their observation corroborates our results (and casts further doubts on the negative findings of Gardner et al., 2017). In another study, Gardner et al., 2019, showed disruption of economic decisions through LO inactivation, albeit using a modified task design. In conclusion, there is clear and converging evidence that LO inactivation disrupts economic choice behavior independent of the sensory modality used to present the offers.

Can the authors further confirm that there were not off target effects of overlying motor cortex, which was also infected with virus (Figure 3G, left)? Were they any changes in response times, number of licks, etc.?

We addressed this issue with an analysis of licking behavior. Please see above, Essential revision points 1 and 2.

Additionally, in the Discussion section, when the authors attempt to reconcile their results with the Gardner paper, they include the following statements: "Second, a close observation of their behavioral data reveals that in the last 3 training sessions, the performance of the experimental group was higher than that of the control group (their Supplementary Figure 2A). If data from these three sessions were combined, the difference between the two groups of rats would presumably be comparable to the difference observed under LO inactivation (their Supplementary Figure 2C)." Out of curiosity I attempted to track this down, but I'm confused. Are they referencing the Gardner et al., 2017 paper? Supplementary Figure 2 in that paper does not have A or C subpanels, and I couldn't make sense of their statement from that figure. Please clarify.

Sorry for this typo and thanks for catching it! It is Figure 6A and Figure 6C of the Gardner study.

Reviewer #2:[…] There are a few areas that the authors could clarify for a more complete description of their findings:1) For readers familiar with the previous work in primates, the description of the rodent task would benefit from greater clarification of details. Describing the liquid reward as 'juices' creates an expectation that mice are choosing between apple juice, grape juice, fruit punch, tea, etc. The methods describe training mice to choose between different quantities of water and 12% sucrose solution. In the rodent literature, these are not typically referred to as juices, but just fluid reward. Here juices A and B are used to refer to the preferred and non-preferred fluid – in two-bottle preference tests, mice typically prefer sucrose. In this paradigm, how often is water the preferred option? As mice are fluid restricted and sucrose solution presents an osmotic challenge, does preference change over the course of a session, or over the course of days of testing?

Thanks for helping us clarify this point. Throughout the experiments, we used the same two liquid rewards, namely water and sucrose water. We refer them as ‘juices’ for uniformity of language with the monkey studies. Throughout the experiments, we labeled sucrose water as ‘juice A’ and water as ‘juice B’. In the vast majority of sessions, sucrose water (juice A) was preferred to water (juice B). This point can be inferred from Figure 2A, showing that in most cases (323/335 = 96%) we measured ρ>1. In the remaining sessions (12/335 = 4%), we measured ρ<1. We clarified these points in the subsection “Economic choice task”.

2) There does not seem to be an introduction of a wider range of juice offers or olfactory cues. Indeed, it would be very surprising if mice (or rats) could flexibly adapt and produce reliable behavior in the more complex version of the task that monkeys perform. I would suggest making it more explicitly clear in the Results that the rodent menu is reduced compared to the monkey and include in the Discussion the experimental advantages to the rodent model for elucidating mechanism as well as the reduced flexibility in the behaviors available to study.

We clarified in the Materials and methods (subsection “Economic choice task”) that only two liquid rewards were used in these experiments. We are reluctant to elaborate on the lack of flexibility in mice because we have not really attempted to use a large number of liquid rewards.

3) Were the spouts continuously available, or did they retract between trials? If spouts are continuously available, behavior between trials may provide another index of spatial biases in the animals.

We clarified in the Materials and methods that the liquid spouts did not retract during odor presentation (subsection “Economic choice task”). Anecdotally, we had actually built the setup with retractable spouts; however, we found that retracting them during the delay increased the electric noise in neuronal recordings.

In general, we found that licking during the delay was very consistent with that after the go cue. In other words, animals simply anticipated the response. Hence, spatial biases are well captured by the measures provided in Figure 2.

4) Licks before the 'go' cue were discarded. Did this 'anticipatory' licking correspond with the spatial encoding of LO neurons?

No. The spatial tuning in LO is defined by the hemisphere. That is, neurons on the right hemisphere encode the value offered on the right, and neurons in the left hemisphere encode the value offered on the left. Since both hemispheres are always present, this tuning cannot correspond to any behavioral spatial bias.

5) In the optogenetic stimulation experiments, the timing of stimulation was set to coincide with odor cue onset. Related to #4, if a mouse were engaged in anticipatory licking, did optogenetic stimulation affect that behavior as well as choice? Was stimulation ever delivered in a different temporal window?

The stimulation started at the beginning of odor presentation and lasted 3.8 s – i.e., throughout the offer period plus 1 s (see Materials and methods, subsection “*Optical stimulation, neuronal recordings, and histology”*). For this revision, we analyzed the licking data. As described in the Revision Summary (point 2), optical stimulation reduced licking in a relatively marginal but statistically significant way. Across sessions and animals, licking dropped by roughly 8%. Most importantly, this motor effect, quantified by LSI, was not correlated with the choice deficit induced by the optical stimulation, quantified by SSI.

Reviewer #3:[…] 1) While successfully training mice to perform this advanced economic decision task is certainly a prize, this behavioral task is quite complex that contains multiple cognitive components and requires several stages of training. This leads to my concern of whether silencing LO specifically impaired economic decision-making, or it might have also disrupted other processes involved in this task. For example, would LO inactivation also disrupts the sensorimotor process, which can be critical for the choice behavior based on odor-reward association? From the data plot in Figure 3, it seems that LO inactivation also disrupted the performance in the easiest trials, where there was only one odor and one rewarded side, and correct responses may only require the odor-reward association. A potential experiment to test this possibility is to inactivate LO in a simpler odor discrimination 2AFC task. If mice's performance is also impaired, then it would be difficult to tell whether LO was only involved in economic decisions or it is generally required for sensorimotor choice behavior. In fact, given the broad projection of OFC neurons and the nonselective nature of silencing methods, it may not be unexpected that the inactivated OFC neurons may also be involved in more general choice processes. If this is the case, the authors may need to confine their interpretation of inactivation results and rely more on the analysis of the recording data to understand the function of OFC neurons.

We agree that these are critical issues, and we have conducted new analyses to address them. The results summarized above (Essential revisions), essentially confirm that the choice deficits induced by optical stimulation are due to inactivation of LO, as opposed other areas such as the olfactory bulb or area M2. We also showed that insofar as the optical stimulation affected licking, this motor effect was not correlated with the induced choice deficits. A more subtle question is whether LO inactivation disrupted the process of value comparison per se (i.e., the decision), as opposed to other aspects of the task – in particular, processing of the olfactory stimulus. Although we cannot completely rule out the latter interpretation based on our data, two lines of evidence argue against it.

First, at the last SfN meeting, Gore et al., 2019, presented a study in which rats performed a juice choice task very similar to ours, except that offers were represented by visual stimuli. Gore and colleagues used optogenetics to inactivate LO and found that this manipulation induced choice deficits very similar to the ones reported here. Their observation corroborates our results. Along similar lines, Gardner et al., 2019 (also presented at the last SfN) showed disruption of economic decisions through LO inactivation, albeit using a modified task design. In conclusion, there is converging evidence that LO inactivation disrupts economic choice behavior independent of the sensory modality. Second, in another recent study, Wu et al., 2019, showed that optogenetic inactivation of area LO – using an approach very similar to ours – did not disrupt performance in an olfactory DMS task. A new paragraph in the Discussion addresses these issues as follows:

“Our choice task relied on olfactory stimuli and licking responses. Since viral injections extended beyond LO, one concern was whether choice deficits induced by optical stimulation were due to inactivation of the olfactory bulb or other olfactory areas. […] Thus, most likely, LO inactivation in our experiments disrupted economic decisions, not olfactory processes.”

2) The authors did not provide sufficient details of virus injection, such as the volume and titer of injected virus, and the injection methods etc, which can influence the range of expression. Although the author optically targeted the LO region, the virus infection regions seem to be quite large (Figure 1G). Photo stimulation may potentially influence neighboring regions. While this issue can be difficult to address experimentally given the difficulty of the task training, the authors should at least discuss the potential caveats.

We added the details on viral injections to the Materials and methods (subsection “Animals and surgical procedures”). In essence, we injected 100 nl of virus (titer = 7*10^12^ vg/ml) using a Hamilton syringe over the course of 10 min, and we waited 2 min before retracting the syringe.

3) The authors found that neuronal activity in different time epochs encode different aspects of the economic decisions. It would be interesting to perform optogenetic silencing restricted in different trial epochs. This would be an optional set of experiments that might not be required to support the current conclusions in this study, but this would provide more informative results to support the specific role of OFC in the economic decisions.

We agree that optogenetic silencing in narrower and properly defined time windows could be extremely interesting. We are currently conducting a couple of follow-up studies, and we will definitely keep in mind this suggestion.

4) It is still very intriguing why LO mainly encode values associated with the ipsilateral side. The authors speculated that this may be specific to the olfactory sensory pathways. This explanation can be better supported if the authors could provide references or evidence supporting that the circuit pathways linking olfactory bulb eventually to OFC are also primarily ipsilateral.

We added the following sentence and references:

“… connections between the olfactory bulb and piriform cortex, and between piriform and orbital cortices are primarily ipsi-lateral (Davis and Macrides, 1981; Datiche and Cattarelli, 1996; Illig, 2005; Hintiryan et al., 2012; Chen et al., 2014)”.

[Editors' note: further revisions were suggested prior to acceptance, as described below.]Overall, the reviewers thought that the authors have addressed most of the previous concerns by additional analysis and revising the manuscript. However, they thought that the issue of regional specificity of optogenetic inhibition should be discussed in a more balanced way. The reviewers agreed that the new analyses using the lasso regression cannot completely rule out the possibility that some, if not all, of the results can be accounted for by unintended inactivation of underlying olfactory areas. After discussion among the reviewers, we agreed that the authors should discuss this caveat explicitly.

We thank the reviewers and the editors for the additional comments and for the favorable decision. We revised the Discussion to include the following paragraph:

“Since viral injections extended beyond LO, one concern was whether choice deficits induced by optical stimulation were due to inactivation of the olfactory bulb or other olfactory areas. […] As a caveat, since the LASSO regression forces most coefficients to zero, one cannot rule out that choice deficits were partly due to inactivation of olfactory regions.”